# Privacy Profiles under Tradeoff Composition

**Paul Glasserman** *pg20@columbia.edu*
*Columbia Business School*
*Columbia University*

**Reviewed on OpenReview:** *https://openreview.net/forum?id=gRvKjXWacu*

## Abstract

Privacy profiles and tradeoff functions are two frameworks for comparing differential privacy guarantees of alternative privacy mechanisms. We study connections between these frameworks. We show that the composition of tradeoff functions corresponds to a binary operation on privacy profiles we call their *T-convolution*. Composition of tradeoff functions characterizes group privacy guarantees, so the T-convolution provides a bridge for translating group privacy properties from one framework to the other. Composition of tradeoff functions has also been used to characterize mechanisms with log-concave additive noise; we derive a corresponding property based on privacy profiles. We also derive new bounds on privacy profiles for log-concave mechanisms based on new convexity properties. In developing these ideas, we characterize *regular* privacy profiles, which are privacy profiles for mutually absolutely continuous probability measures.

## 1 Introduction

Differential privacy (DP, introduced in Dwork et al. (2006b)) is a widely used framework for quantifying the efficacy of mechanisms for protecting data privacy. The framework applies in particular to mechanisms that add noise to data, usually with the goal of balancing access to data and privacy protection. DP seeks to ensure that an attacker is unlikely to be able to detect small differences between datasets that reveal individual values. In $(\epsilon, \delta)$-DP, the parameters $\epsilon$ and $\delta$ control a mechanism's security by limiting relative and absolute differences in the probabilities of a mechanism's response when applied to close but distinct datasets.

Privacy profiles (introduced in Balle et al. (2020)) and tradeoff functions (introduced in Dong et al. (2022)) are tools for representing and analyzing DP privacy guarantees. A privacy profile takes one DP parameter ($\delta$) as a function of the other ($\epsilon$) and can be viewed as menu of parameter combinations offered by a mechanism. For a fixed mechanism, requiring a more stringent (smaller) value of $\epsilon$ requires accepting a looser (larger) guarantee for $\delta$. Comparing privacy profiles allows a comparison of the parameter combinations offered by alternative mechanisms.

Tradeoff functions, in the sense of Dong et al. (2022), are grounded in a hypothesis testing view of data privacy. Consider an attacker trying to determine whether a query response comes from one database or another. Think of the two databases as differing only in the value of a single entry, whose privacy needs to be protected. The attacker's null and alternative hypotheses state that the source is one database or the other. In this setting, privacy protection means making the hypothesis test difficult for the attacker by making query responses from the two databases difficult to distinguish. Any privacy mechanism introduces type I and type II errors for the attacker; a tradeoff function determines the lowest probability of one error for each probability of the other. Indeed, the tradeoff function is closely related to the receiver operating characteristic of hypothesis testing. The privacy guarantees of alternative mechanisms can be compared through their tradeoff functions.

The goal of this paper is to develop novel connections between privacy profiles and tradeoff functions and to use these connections to derive bounds on privacy guarantees. Dong et al. (2022) show that tradeoff

functions and $(\epsilon, \delta)$-DP guarantees are related through convex conjugation; Zhu et al. (2022) extend this to establish a bijection between privacy profiles and tradeoff functions. We build on these connections.

Our main result introduces to the DP setting a binary operation on privacy profiles we call the *T-convolution*. The T-convolution of a pair of privacy profiles yields another profile; we show that the resulting profile corresponds to the composition of the corresponding tradeoff functions. In other words, the T-convolution of privacy profiles corresponds to the composition of their tradeoff functions (which should not be confused with the composition of mechanisms). The composition of tradeoff functions is important because it characterizes *group privacy* guarantees implied by a mechansim (as in Dong et al. (2022), Theorem 3); group privacy is concerned with preserving privacy between databases that differ in a group of values, as opposed to just in an individual value. The T-convolution thus provides a bridge between privacy profiles and tradeoff functions that translates statements about group privacy in one framework into the other.

In developing these ideas, we introduce the notion of *regular* privacy profiles. We show that the privacy profile for a pair of probability measures is regular if and only if the probability measures are mutually absolutely continuous. A failure of absolute continuity is a failure of privacy because it implies that some of a mechanism's responses perfectly identify the source of the response. Zhu et al. (2022) characterize privacy profiles as convex functions satisfying some additional properties; we show that regular privacy profiles are characterized by two additional boundary conditions. We also show that the T-convolution of regular privacy profiles is regular. This is convenient because privacy protection argues for limiting consideration to mechanisms with regular profiles.

In addition to its significance for group privacy, tradeoff function composition is central to the notion of infinite divisibility introduced in Awan & Dong (2022). Awan & Dong (2022) use infinite divisibility to characterize additive noise mechanisms in which the noise has a log-concave density, a family that includes the Gaussian and Laplace densities. The T-convolution provides a corresponding notion of infinite divisibility of privacy profiles and thus a characterization of privacy profiles for additive log-concave noise.

In the last part of the paper, we study log-concave privacy profiles in greater depth. A log-concave profile has a *sensitivity* parameter $y$ that reflects differences in true query responses (before noise is added) between databases. We show that the T-convolution of log-concave privacy profiles takes a particularly simple form — it corresponds to the addition of sensitivity parameters. This leads to bounds on log-concave profiles.

We also study convexity properties of log-concave profiles with respect to the sensitivity parameter $y$. We show that a log-concave profile is sigmoidal, meaning that it is convex at small values of $y$ and concave at larger values of $y$, with an explicitly identified inflection point. In the convex region, privacy loss accelerates as the sensitivity parameter increases; the opposite holds in the concave region. We also establish a *relative* convexity property, which we may paraphrase as saying that at a larger value of $\epsilon$, privacy loss (from increasing sensitivity $y$) accelerates more quickly.

We formulate most of our results on privacy profiles for a pair of probability measures. As in Kairouz et al. (2015), Su (2024), and Wasserman & Zhou (2010), this corresponds to a binary hypothesis test between two databases. More generally, through the notion of a dominating pair of probability measures introduced by Zhu et al. (2022), the privacy profile for a mechanism applied to multiple databases can often be reduced to the privacy profile for a worst-case pair of measures, to which our results can then be applied.

Because we formulate most of our results in terms of generic probability measures, most of our results are not restricted to scalar mechanisms or mechanisms of a particular form. We do limit ourselves to scalar mechanisms when we discuss additive noise and, in particular, log-concave additive noise. These results could be extended to multidimensional noise with independent and identically distributed log-concave components through Theorem 9 of Vinterbo (2022).

Some of the ideas we use in this paper have counterparts in the financial mathematics literature. The starting point for this connection is the fact that the "hockey-stick divergence" that underpins privacy profiles has the form of a call option in the financial setting. The privacy profile for the Gaussian mechanism, first derived in Balle & Wang (2018), coincides with the Black & Scholes (1973) formula central to option pricing. What we call the T-convolution was introduced to the option pricing literature by Tehranchi (2020). The characterization of symmetry for privacy profiles is closely related to the notion of put-call symmetry

discussed in Carr & Lee (2009). The connection between infinite divisibility of tradeoff functions and log-concave additive noise has a counterpart in the actuarial science literature (Cherny & Filipović (2011)). We note these parallels where they are relevant, particularly in cases where the interpretation of similar properties across the two problem domains is far from obvious.

To summarize, the paper's main contributions are the following:

- We show that composition of tradeoff functions corresponds to the T-convolution of privacy profiles, thus linking properties of group privacy and log-concave mechanisms across the two frameworks.

- We characterize the privacy profiles of mutually absolutely continuous probability measures as *regular* privacy profiles, defined by two simple conditions, with counterparts for tradeoff functions. We also characterize symmetric privacy profiles.

- We derive novel convexity properties for the privacy profiles of log-concave mechanisms, leading to new bounds.

Section 2 provides background on privacy profiles and tradeoff functions. Section 3 characterizes regular privacy profiles. Section 4 defines the T-convolution and studies its properties and its implications for group privacy and log-concave mechanisms. Section 5 presents new convexity properties for log-concave mechanisms. Section 6 concludes, and most proofs are deferred to an appendix.

## 2 Preliminaries: Privacy Profiles and Tradeoff Functions

We are given a collection $\mathcal{X}$ of databases and a symmetric binary relation $\simeq$ for which we interpret $x \simeq x'$ to mean that $x, x' \in \mathcal{X}$ are neighboring databases. For example, we might take $x \simeq x'$ to mean that $x$ and $x'$ differ in one entry. A mechanism $M$ is a randomized algorithm on $\mathcal{X}$; for each $x \in \mathcal{X}$, $M(x)$ is a random variable or vector representing the response returned by the mechanism when applied to $x$. We associate with $x \in \mathcal{X}$ a probability measure $P$, with the interpretation that $P(A)$ is the probability that $M(x) \in A$, for any (suitably measurable) set $A$. We express this as $M(x) \sim P$.

**Definition 1.** *(Dwork et al., 2006b),Dwork et al. (2006a)) The mechanism $M$ on $(\mathcal{X}, \simeq)$ satisfies $(\epsilon, \delta)$-differential privacy (DP), for some $\epsilon \geq 0$ and $0 \leq \delta \leq 1$, if, for all measurable sets $A$,*

$$Q(A) \leq e^{\epsilon} P(A) + \delta, \tag{1}$$

*whenever $M(x) \sim P$ and $M(x') \sim Q$ for some $x, x' \in \mathcal{X}$ with $x \simeq x'$.*

Balle et al. (2020) define a privacy profile (a privacy curve in Gopi et al. (2021)) for a mechanism, based on which we introduce the following essentially equivalent definition.

**Definition 2.** *For any two probability measures $P$ and $Q$ on the same measurable space, the privacy profile $\delta_{P,Q} : [0, \infty) \to [0, 1]$ is the function*

$$\delta_{P,Q}(K) = \sup_A \{Q(A) - KP(A)\}, \tag{2}$$

*the supremum taken over measurable sets. The privacy profile for a mechanism $M$ is the supremum of $\delta_{P,Q}(K)$ over all $(P, Q)$ with $M(x) \sim P$ and $M(x') \sim Q$ for some $x, x' \in \mathcal{X}$ with $x \simeq x'$.*

We will simplify $\delta_{P,Q}$ to $\delta$ when $Q$ and $P$ are fixed. To connect (2) to (1), take $K = e^{\epsilon}$. It will be convenient to consider all $K \geq 0$, even though requiring $\epsilon \geq 0$ would limit us to $K \geq 1$. *Except where otherwise noted, we take the privacy profile to be a function of $K \geq 0$ rather than $\epsilon$.*

Balle et al. (2020) use a different but equivalent definition based on the following "hockey-stick divergence." As usual, we write $Q \ll P$ if $Q$ is absolutely continuous with respect to $P$, and we write $Q \ll\gg P$ if $P$ and $Q$ are mutually absolutely continuous. We write $x^+$ for $\max\{x, 0\}$.

**Definition 3.** *For probability measures $P$ and $Q$ on the same measurable space with $Q \ll P$, and any $K \geq 0$,*

$$H_K(Q|P) = \mathbb{E}_P\left[\left(\frac{dQ}{dP} - K\right)^+\right]. \tag{3}$$

Balle et al. (2020) define the privacy profile of a mechanism $M$ to be the supremum of $H_K(Q|P)$ over all $P \sim M(x)$ and $Q \sim M(x')$ for neighboring databases $x, x'$. The definitions are essentially equivalent because if $Q \ll P$, then $\delta_{P,Q}(K) = H_K(Q|P)$; see Theorem 1 of Balle et al. (2020) and references there, particularly Proposition 2 of Barthe & Olmedo (2013).[1]

If $Q$ fails to be absolutely continuous with respect to $P$, then the calculation of the divergence $H_K(Q|P)$ involves an additional term consistent with general properties of divergence measures. As explained in Ali & Silvey (1966), pp.133-134, there is a set $A$, with $P(A) = 1$, such that $Q \ll P$ when $Q$ and $P$ are restricted to $A$, and for which $Q(A^c)$ is largest, where $A^c$ is the complement of $A$. Informally, $A^c$ is the set of outcomes that are possible under $Q$ but not under $P$. (See Ali & Silvey (1966), p.133, for a precise definition; our set $A^c$ is their set $N$.) Then (3) generalizes to

$$H_K(Q|P) = \mathbb{E}_P\left[\left(\frac{dQ}{dP} - K\right)^+ \mathbf{1}_A\right] + Q(A^c). \tag{4}$$

If, for example, we take $K = 0$, then (4) yields $H_K(Q|P) = Q(A) + Q(A^c) = 1$, consistent with what we get from (2), even when $Q \not\ll P$. From now on, we take $\delta_{P,Q}(K) = H_K(Q|P)$, with the understanding that $H_K(Q|P)$ takes the form in (4) when $Q$ fails to be absolutely continuous with respect to $P$.

We turn next to the definition of tradeoff functions as introduced in Dong et al. (2022) and modified in Awan & Dong (2022). (The following definition corrects a minor typo in Awan & Dong (2022), also noted in Awan & Ramasethu (2024).)

**Definition 4.** *For any two probability measures $P$ and $Q$ on the same measurable space, the tradeoff function $T(P,Q) : [0,1] \rightarrow [0,1]$ is the function*

$$T(P,Q)(\alpha) = \inf_\phi\{1 - \mathbb{E}_Q[\phi] : \mathbb{E}_P[\phi] \leq 1 - \alpha\}, \tag{5}$$

*the infimum taken over all measurable rejection rules (i.e., random variables $\phi$ with $0 \leq \phi \leq 1$).*

Suppose $M(x) \sim P$ and $M(x') \sim Q$. Think of $\phi$ as a function of the mechanism's output used by an adversary to infer whether the mechanism was applied to database $x$ or $x'$. The adversary's decision rule is to infer with probability $\phi$ that the true database is $x'$ and $x$ otherwise. The tradeoff function traces the optimal type II error over such rules at each value of one minus the type I error. The original definition in Dong et al. (2022) is

$$T_o(\alpha) = T(1 - \alpha). \tag{6}$$

Proposition 1 of Dong et al. (2022) shows that $T_o : [0,1] \rightarrow [0,1]$ is a tradeoff function (in their original definition) if and only if $T_o$ is continuous, convex, and decreasing, with $T_o(x) \leq 1 - x$, for all $x \in [0,1]$. (Here and throughout, we use "increasing" and "decreasing" in their weak sense and insert "strictly" where the strict sense is intended.) Thus, under the definition of Awan & Dong (2022), $T : [0,1] \rightarrow [0,1]$ is a tradeoff function if and only if

$$T \text{ is continuous, convex, and increasing, with } T(x) \leq x, \text{ for all } x \in [0,1]. \tag{7}$$

Zhu et al. (2022) derive a related characterization of hockey-stick divergences and thus, implicitly, of privacy profiles through the set of functions $\delta : \mathbb{R}_+ \rightarrow [0,1]$ satisfying

$$\delta \text{ is convex, decreasing, with } \delta(0) = 1 \text{ and } \delta(K) \geq (1 - K)^+, \text{ for all } K \geq 0. \tag{8}$$

Note that the condition $\delta(0) = 1$ requires the representation in (4) if $Q \not\ll P$.

---

[1]What we call $\delta_{P,Q}$ is sometimes called $\delta_{Q,P}$. Our notational preference anticipates the connection between privacy profiles and tradeoff functions in Section 4.1. With our notation, $\delta_{P,Q}$ is linked to the tradeoff function for $(P,Q)$ rather than $(Q,P)$.

## 3 Regular Privacy Profiles

### 3.1 Definition and Conditions for Regularity

A failure of absolute continuity is a failure of privacy: if $M(x)$ can return values that are impossible under $M(x')$, then observing such a value allows an adversary to definitively distinguish between the two databases. We are therefore interested in characterizing privacy profiles $\delta_{P,Q}$ of mutually absolutely continuous probability measures. In other words, of all the functions $\delta$ satisfying the conditions in (8), how can we distinguish the ones that are privacy profiles for some mutually absolutely continuous probability measures? We will call these *regular* privacy profiles. Besides being of independent interest, we will need this characterization for later results.

As a bounded and decreasing function, $\delta(K)$ has a limit as $K \to \infty$ which we denote by $\delta(\infty)$. As a convex function, any $\delta$ satisfying (8) has at least a right-derivative $\delta'(K)$ at all $K \geq 0$. (For this and other properties of convex functions, see Appendix A.) This holds even if the likelihood ratio $dQ/dP$ is discrete-valued. (It will follow from the representation in Lemma 1, below, that if the likelihood ratio is discrete-valued, then $\delta'(K)$ is piecewise constant, so $\delta(K)$ is piecewise linear.)

**Definition 5.** *A function $\delta : \mathbb{R}_+ \to [0,1]$ satisfying (8) is* regular *if it also satisfies $\delta'(0) = -1$ and $\delta(\infty) = 0$.*

Zhu et al. (2022), Lemma 9, implies that a function $\delta$ is the privacy profile for a pair of probability measures if and only if it satisfies (8). We extend their result as follows:

**Proposition 1.** *A function $\delta$ is the privacy profile of a pair of mutually absolutely continuous probability measures if and only if it is regular.*

The interpretation of this result will be as follows: the condition $\delta'_{P,Q}(0) = -1$ ensures that $dQ/dP$ does not have mass at zero, and the condition $\delta_{P,Q}(\infty) = 0$ ensures that $dQ/dP$ does not have mass at infinity.

In developing this connection, the proof of the proposition uses the following lemma, which characterizes the distribution of a likelihood ratio through a regular privacy profile. By extracting the distribution of the likelihood ratio from a privacy profile, we can determine whether the profile is compatible with a mutually absolutely continuous pair of probability measures: the likelihood ratio should be strictly positive and finite.

**Lemma 1.** *Suppose $\delta$ is a regular privacy profile, and define $G(x) = 1 + \delta'(x)$, $x \geq 0$. Then $G$ is a cumulative distribution on $\mathbb{R}_+$, with $G(0) = 0$. If the random variable $L$ has distribution $G$ under a probability measure $P$, then*

$$\mathbb{E}_P[(L-K)^+] = \int_K^\infty P(L > x)\,dx = \int_K^\infty [1 - G(x)]\,dx = \delta(K), \quad K \geq 0. \tag{9}$$

As a consequence of this result (see the proof in Appendix B), we have that if $P \ll\gg Q$, then, for all $x \in \mathbb{R}$,

$$P\left(\frac{dQ}{dP} \leq x\right) = 1 + \delta'_{P,Q}(x) =: G_{P,Q}(x); \tag{10}$$

in particular, the privacy profile $\delta_{P,Q}$ determines the distribution $G_{P,Q}$ of the likelihood ratio $dQ/dP$.

The random variable $-\log(dQ/dP)$ is often called the privacy loss random variable, so the privacy loss distribution (under $P$) is $1 - G_{P,Q}(e^{-x}) = -\delta'_{P,Q}(e^{-x})$. Thus, Lemma 1 also connects privacy profiles with the privacy loss distribution used by Dwork & Rothblum (2016), Sommer et al. (2019), Koskela et al. (2020), and Gopi et al. (2021), among many others.

We illustrate Lemma 1 and Proposition 1 with examples.

**Example:** (Gaussian mechanism). Consider a mechanism that adds a standard normal random variable $Z$ to a query response. Consider two databases with query values $y_0$ and $y_0 + y$, $y > 0$. Let $P$ and $Q$ be the probability measures associated with the query responses $y_0 + Z$ and $y_0 + y + Z$. It follows from Theorem 8 of Balle & Wang (2018) that the privacy profile $\delta_{P,Q}$ is given by

$$\delta(K) = \Phi\left(-\frac{\log K}{y} + \frac{y}{2}\right) - K\Phi\left(-\frac{\log K}{y} - \frac{y}{2}\right), \tag{11}$$

where $\Phi$ is the standard normal cumulative distribution function. The measures satisfy $P \ll\gg Q$, and (11) is easily seen to be regular. The cdf $G$ introduced in Lemma 1 evaluates to

$$1 + \delta'(x) = \Phi\left(\frac{\log x}{y} + \frac{y}{2}\right) = P\left(e^{yZ - y^2/2} \le x\right) = P\left(\frac{dQ}{dP} \le x\right), \tag{12}$$

which confirms that $1 + \delta'$ is indeed the cdf of the likelihood ratio. This analysis extends to multiple databases if we take $y$ to be the largest difference in query responses across all pairs of databases, called the sensitivity of the query. If we replace $Z$ with $\sigma Z$, for some $\sigma > 0$, then $y$ in (11) becomes $y/\sigma$, which amounts to a change of units. We will refer to $y$ as the sensitivity parameter (or the scaled sensitivity parameter) for the mechanism. (The expression in (11) coincides with the Black & Scholes (1973) formula for the price of a call option with a strike price of $K$ and an interest rate of zero. In that setting $y$ is the total volatility to expiration. In both settings, (11) arises as the expectation of the function $x \mapsto (x - K)^+$ applied to a lognormal random variable.)

**Example:** (Randomized response). Balle et al. (2020) consider a mechanism acting on $\{0, 1\}$. The mechanism flips each bit with probability $1 - p$, $p \in [1/2, 1)$, and leaves it unchanged with probability $p$. Let $P$ and $Q$ be the probability measures on $\{0, 1\}$ describing the distribution of the mechanism applied to 0 and 1, respectively, so $P(0) = Q(1) = p$ and $P(1) = Q(0) = 1 - p$. Balle et al. (2020) derive the corresponding privacy profile

$$\delta(e^\epsilon) = [p - e^\epsilon(1 - p)]^+, \quad \epsilon \ge 0.$$

Writing $K = e^\epsilon$ and right-differentiating with respect to $K$ at $K = 0$ yields $\delta'(K = 0) = -(1 - p) \ne -1$, which would appear to violate regularity, despite the fact that $P \ll\gg Q$. The apparent contradiction is resolved by noting that when we extend $\delta(K)$ to all $K \in [0, \infty)$ (rather than just $K \ge 1$), following the same argument as in Balle et al. (2020), we get

$$\delta(K) = [(1 - p) - Kp]^+ + [p - K(1 - p)]^+, \tag{13}$$

and $\delta'(0) = -p - (1 - p) = -1$, as required for regularity. Thus, this example illustrates a general point: checking the regularity condition at $K = 0$ requires defining the privacy profile on $[0, \infty)$, even though differential privacy is usually concerned only with $K \ge 1$ (i.e., $\epsilon \ge 0$).

**Dominating pair of distributions.** Zhu et al. (2022) call $(Q, P)$ a tightly dominating pair of distributions for a mechanism $M$ if, for all $K \ge 0$,

$$\sup_{x \simeq x'} H_K(M(x')|M(x)) = H_K(Q|P). \tag{14}$$

Proposition 8 of Zhu et al. (2022) states that a tightly dominating pair always exists. We can add to this property that if $M(x) \ll\gg M(x')$, for all $x \simeq x'$ in a finite collection of databases, then a tightly dominating pair with $P \ll\gg Q$ exists. The proof is as follows. Every $\delta_{M(x'), M(x)}$ is regular. The properties characterizing regular privacy profiles are preserved by taking the maximum over a finite set of profiles, so $\delta(K) = \max_{x \simeq x'} \delta_{M(x'), M(x)}(K)$ is a regular privacy profile. As in the proof of Proposition 1, $G(K) = 1 - \delta'(K)$ defines the cdf of $dQ/dP$ under $P$; any pair of distributions for which $dQ/dP$ has distribution $G$ provides a tightly dominating pair. For example, one may take $P$ to be uniform on $(0, 1)$ and then define $Q$ to have the cdf $G^{-1}(\cdot)$ on $(0, 1)$. (This recovers the construction in Zhu et al. (2022).) This is just one of many possible constructions. The more fundamental quantity is $G$, the distribution of the likelihood ratio for *any* tightly dominating pair.

## 3.2 Symmetric Privacy Profiles

The definition in (14) is symmetric in $x$ and $x'$, which suggests that we should consider suitably symmetric privacy profiles. A natural requirement for symmetry of a privacy profile $\delta_{P,Q}$ is the condition $\delta_{P,Q} = \delta_{Q,P}$. But if we are given a function $\delta$ satisfying the profile conditions (8) but not specifically tied to a pair $(P, Q)$, what does it mean for $\delta$ to be symmetric? This question is answered through the transformation

$$\hat{\delta}(K) = 1 - K + K\delta(1/K), \quad K \ge 0, \tag{15}$$

where, as before, $\delta(\infty)$ should be interpreted as the limit of the decreasing, nonnegative function $\delta(\cdot)$ at $\infty$. We will call a privacy profile $\delta$ symmetric if $\delta = \hat{\delta}$. The transformation (15) combines two transformations that are used in the option pricing literature: $K \to K\delta(1/K)$ arises in put-call symmetry, and $t \to 1 - K + t$ arises in put-call parity; see, e.g., Carr & Lee (2009). The combination of the two turns one hockey-stick divergence (call option) into another. In our setting, this transformation has the following properties, which justify our definition of symmetry.

**Proposition 2.** *With the operation in (15),*

*(i)* $\delta_{Q,P} = \hat{\delta}_{P,Q}$.

*(ii) If $\delta$ is a privacy profile, then so is $\hat{\delta}$, and $\hat{\hat{\delta}} = \delta$.*

*(iii) If $\delta$ is regular, then so is $\hat{\delta}$.*

Property (i) also follows from Lemma 46 of Zhu et al. (2022). Property (iii) will be useful in verifying regularity. As we show in the proof of the proposition,

$$\delta'(0) = -1 \Leftrightarrow \hat{\delta}(\infty) = 0. \tag{16}$$

In other words, the operation in (15) interchanges the two conditions for regularity, so we can verify one boundary condition for $\delta$ by verifying the opposite boundary condition for $\hat{\delta}$. This property makes sense in light of part (i) of the proposition because ruling out a mass at zero (or infinity) for $dQ/dP$ is equivalent to ruling out a mass at infinity (or zero) for $dP/dQ$.

## 4 Composition and Convolution

Dong et al. (2022) show that group privacy can be characterized through the composition of tradeoff functions. Tradeoff composition is also used in Awan & Dong (2022) to define "infinite divisibility," which characterizes the important class of additive log-concave noise mechanisms, which includes the Laplace and Gaussian mechanisms, among many others. The main result of this section (Theorem 1) establishes a correspondence between composition of tradeoff functions and a binary operation on privacy profiles.

### 4.1 Linking Tradeoff Functions and Privacy Profiles

We first recall a link between tradeoff functions and privacy profiles introduced in Dong et al. (2022), Proposition 6, through convex conjugacy. For a function $g : \mathbb{R} \to \mathbb{R}$, the convex conjugate $g^* : \mathbb{R} \to \mathbb{R}$ is defined by setting

$$g^*(y) = \sup_{x \in \mathbb{R}} \{yx - g(x)\}. \tag{17}$$

For $g$ defined on a subinterval of $\mathbb{R}$, the supremum is taken over that subinterval; equivalently, one may extend $g$ to all of $\mathbb{R}$ by setting $g(x) = \infty$ outside its domain. (See Appendix A for additional background.) Translating Proposition 6 of Dong et al. (2022) and Lemma 20 of Zhu et al. (2022) to our setting yields the following:

**Lemma 2.** *Let $T$ be the tradeoff function and $\delta$ the privacy profile for $(P, Q)$. Then*

$$\delta(K) = 1 - K + T^*(K), \quad K \geq 0, \tag{18}$$

*and*

$$T(\alpha) = 1 + \delta^*(\alpha - 1), \quad 0 \leq \alpha \leq 1. \tag{19}$$

*Proof.* Lemma 20 of Zhu et al. (2022) says that $\delta(K) = 1 + T_o^*(-K)$, for all $K \geq 0$, with $\delta$ the privacy profile for $(P, Q)$ and $T_o$ the tradeoff function (under the definition in (6)) for the same $(P, Q)$; we include a proof

of this result in Lemma 10 of Appendix C. Using the definition of the convex conjugate $T_o^*$, we get

$$
\begin{aligned}
T_o^*(-K) &= \sup_{0 \leq \alpha \leq 1} \{-K\alpha - T_o(\alpha)\} \\
&= -K + \sup_{0 \leq \alpha \leq 1} \{K(1-\alpha) - T_o(\alpha)\} \\
&= -K + \sup_{0 \leq \alpha \leq 1} \{K\alpha - T_o(1-\alpha)\} \\
&= -K + \sup_{0 \leq \alpha \leq 1} \{K\alpha - T(\alpha)\} \\
&= -K + T^*(K),
\end{aligned}
$$

so (18) follows. For (19), we use the fact (see Appendix A) that $T^{**} = T$. Thus,

$$
\begin{aligned}
T(\alpha) = T^{**}(\alpha) &= \sup_{K \in \mathbb{R}} \{\alpha K - T^*(K)\} \\
&= \sup_{K \in \mathbb{R}} \{\alpha K - [\delta(K) + K - 1]\} \\
&= 1 + \sup_{K \in \mathbb{R}} \{(\alpha - 1)K - \delta(K)\} \\
&= 1 + \delta^*(\alpha - 1).
\end{aligned}
$$

$\square$

In light of Lemma 2, for a privacy profile $\delta$ we let

$$
T_\delta(\alpha) = 1 + \delta^*(\alpha - 1), \quad 0 \leq \alpha \leq 1, \tag{20}
$$

denote the tradeoff function associated with $\delta$, and for a tradeoff function $T$ we let

$$
\delta_T(K) = 1 - K + T^*(K), \quad K \geq 0, \tag{21}
$$

denote the privacy profile associated with $T$. These operations are inverses of each other. For example, if we map $\delta \to T_\delta \to \delta_{T_\delta}$, we get the original $\delta$ because

$$
\begin{aligned}
\delta_{T_\delta}(K) &= 1 - K + T_\delta^*(K) \\
&= 1 - K + \sup_\alpha \{\alpha K - T_\delta(\alpha)\} \\
&= 1 + \sup_\alpha \{(\alpha - 1)K - [1 + \delta^*(\alpha - 1)]\} \\
&= \sup_\alpha \{\alpha K - \delta^*(\alpha)\} \\
&= \delta^{**}(K) = \delta(K). \tag{22}
\end{aligned}
$$

The last step follows from the fact the privacy profile functions are convex, continuous, and finite. (See Appendix A.)

These connections allow us to characterize regular privacy profiles through their associated tradeoff functions:

**Proposition 3.** *A privacy profile $\delta$ is regular if and only if $T_\delta$ satisfies $T_\delta(1) = 1$ and $T_\delta(\alpha) > 0$, for all $\alpha > 0$.*

Figure 1 illustrates Proposition 3. The privacy profile on the right fails to be regular because $\delta'(0) > -1$ and $\delta(\infty) > 0$. (Either property alone would violate regularity.) The associated tradeoff function $T_\delta$ on the left violates the corresponding conditions because it is flat in an interval to the right of $\alpha = 0$ and it fails to reach 1 at $\alpha = 1$.

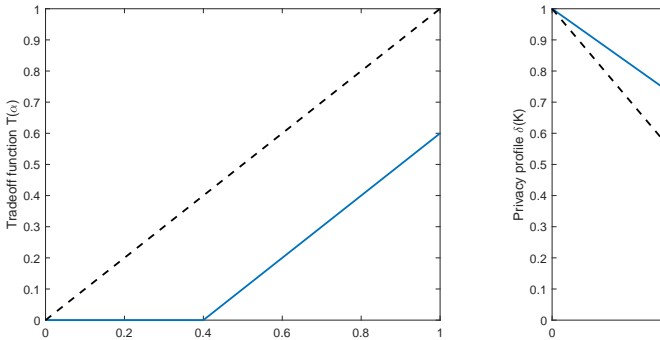

Figure 1: On the left, the tradeoff function $T(\alpha) = (\alpha - p)^+$, $p \in (0,1)$, stays at zero for $\alpha \in (0, p)$ and has a maximum value of $1 - p < 1$. On the right, the corresponding privacy profile $\delta(K) = \max\{1 - (1-p)K, p\}$ therefore has $\delta'(0) = -1 + p > -1$ and $\delta(\infty) = p > 0$, so it fails both conditions for regularity.

## 4.2 The T-Convolution of Privacy Profiles

We now introduce an operation on privacy profiles we call the *T-convolution*, in reference to Tehranchi (2020), who introduced the operation for option prices and derived several properties we apply here. We will see that the T-convolution of privacy profiles corresponds to the composition of tradeoff functions, so the "T" is suggestive in this sense as well.

**Definition 6.** *For privacy profiles $\delta_1, \delta_2$, the T-convolution is given by*

$$\delta_1 \bullet \delta_2(K) = \inf_{\eta \geq 0} \{\delta_1(\eta) + \eta \delta_2(K/\eta)\}, \quad K \geq 0, \tag{23}$$

*with the convention that $0 \cdot \delta_2(K/0) = 0$.*

Before developing the connection with tradeoff functions, we record some properties of this operation on privacy profiles, analogous to properties in Tehranchi (2020) for option prices. Recall the definition of $\hat{\delta}$ in (15).

**Lemma 3.** *For the transformation in (15), we have $\widehat{\delta_1 \bullet \delta_2} = \hat{\delta}_2 \bullet \hat{\delta}_1$.*

**Lemma 4.** *(i) If $\delta_1, \delta_2$ are privacy profile functions, then so is $\delta_1 \bullet \delta_2$. (ii) If $\delta_1$ and $\delta_2$ are regular, then so is $\delta_1 \bullet \delta_2$. (iii) In either case, for each $K > 0$, the infimum over $\eta$ is attained in $[0, K+1]$.*

Our interest in this operation stems from the next result, which shows that composition of tradeoff functions corresponds to T-convolution of privacy profiles. Tehranchi (2020) derives a corresponding result for option prices and notes a related result in Borwein & Vanderwerff (2010). In the following, $\circ$ denotes ordinary function composition: $T_1 \circ T_2(\alpha) = T_1(T_2(\alpha))$. Recall the definition of $T_\delta$ in (20).

**Theorem 1.** *For any privacy profile functions $\delta_1$, $\delta_2$,*

$$T_{\delta_1} \circ T_{\delta_2} = T_{\delta_1 \bullet \delta_2}.$$

*Proof.* Applying (20) twice and then simplifying, we get

$$
\begin{aligned}
T_{\delta_1} \circ T_{\delta_2}(\alpha) &= 1 + \delta_1^*(T_{\delta_2}(\alpha) - 1) \\
&= 1 + \delta_1^*([1 + \delta_2^*(\alpha - 1)] - 1) \\
&= 1 + \delta_1^*(\delta_2^*(\alpha - 1)) \\
&= 1 + \delta_1^*(\sup_{K \geq 0}\{(\alpha - 1)K - \delta_2(K)\}) \\
&= 1 + \sup_{\eta \geq 0}\{\eta \sup_{K \geq 0}\{(\alpha - 1)K - \delta_2(K)\} - \delta_1(\eta)\} \\
&= 1 + \sup_{K \geq 0}\sup_{\eta \geq 0}\{\eta\{(\alpha - 1)K - \delta_2(K)\} - \delta_1(\eta)\} \\
&= 1 + \sup_{K' \geq 0}\sup_{\eta \geq 0}\{(\alpha - 1)K' - \eta\delta_2(K'/\eta) - \delta_1(\eta)\} \\
&= 1 + \sup_{K' \geq 0}\{(\alpha - 1)K' - \inf_{\eta \geq 0}\{\eta\delta_2(K'/\eta) + \delta_1(\eta)\}\} \\
&= 1 + \sup_{K' \geq 0}\{(\alpha - 1)K' - \delta_1 \bullet \delta_2(K')\} \\
&= 1 + (\delta_1 \bullet \delta_2)^*(\alpha - 1) \\
&= T_{\delta_1 \bullet \delta_2}(\alpha).
\end{aligned}
$$

$\square$

The composition of tradeoff functions $T_1 \circ T_2$ is central to determining group privacy (as in Section 2.5 of Dong et al. (2022)) and to the notion of infinite divisibility in Awan & Dong (2022), which characterizes additive mechanisms with log-concave noise, topics to which we return in Sections 4.3 and 4.4. Theorem 1 shows that composition of tradeoff functions corresponds to T-convolution of privacy profiles.

We should emphasize that Theorem 1 refers to the composition of tradeoff functions and not to the composition of mechanisms. Dong et al. (2022), Section 3, and Gopi et al. (2021), Section 2, discuss operations on tradeoff functions and privacy curves that correspond to composition of mechanisms.

For fixed $K$, computation of (23) reduces to a one-dimensional convex minimization problem and can thus be carried out fairly efficiently, at least if $\delta_1$ and $\delta_2$ are easy to evaluate. We do not know of a fast way to calculate (23) for all $K$. In some cases, it can be evaluated explicitly, as we now illustrate.

**Example:** (Gaussian mechanism). The Gaussian privacy profile (11) implicitly depends on the sensitivity parameter $y$. To illustrate the effect of the T-convolution, we make this dependence explicit and write (11) as $\delta(K, y)$. For any $y_1, y_2 \geq 0$, we now have

$$
\delta(K, y_1) \bullet \delta(K, y_2) = \delta(K, y_1 + y_2), \tag{24}
$$

so T-convolution corresponds to addition of sensitivity parameters in the Gaussian mechanism. We will revisit this identity as a consequence of a more general result for log-concave additive noise mechanisms in Section 4.4. When translated to tradeoff functions through Theorem 1, this result recovers Theorem 3 of Dong et al. (2022), which shows that composition of Gaussian tradeoff functions corresponds to addition of sensitivity parameters. For the operation in (24), the minimizing $\eta$ is given by

$$
\eta^* = K^{y_1/(y_1+y_2)}e^{-(y_1+y_2)/2}.
$$

**Example:** (Randomized response). Suppose the privacy profiles $\delta_i$ have the form in (13) with parameters $p_i$, $i = 1, 2$. We can write these profiles in the equivalent form

$$
\delta_i(K) = \max\{1 - K, p_i - K(1 - p_i), 0\}, \quad i = 1, 2,
$$

which yields

$$
\delta_1(\eta) + \eta\delta_2(K/\eta) = \max\{1 - \eta, p_1 - \eta(1 - p_1), 0\} + \max\{\eta - K, p_2\eta - K(1 - p_2), 0\}. \tag{25}
$$

For each $K \geq 0$, this is a piecewise linear function of $\eta$ with break points contained in the set

$$\{Kp_2/(1-p_2), (1-p_1)/p_1, K(1-p_2)/p_2, p_1/(1-p_1)\}.$$

On each $\eta$ interval on which $\delta_1(\eta) + \eta\delta(K/\eta)$ is linear, its minimum is attained at an endpoint of the interval. In minimizing (25) over $\eta$, it therefore suffices to consider $\eta$ values in the breakpoints. In other words, $\delta_1 \bullet \delta_2(K)$ is the minimum of (25) over the four $\eta$ break points. The result can be written in the form

$$\delta_1 \bullet \delta_2(K) = \max\{1-K, 1-q-(1-p_2)K, p_1 - p_1qK/p_2, 0\},$$

with $q = (1-p_1)(1-p_2)/p_1$. More generally, it is easy to see that if $\delta_1$ and $\delta_2$ are piecewise linear, then so is $\delta_1 \bullet \delta_2$. Even if we are ultimately interested in $\delta_1 \bullet \delta_2(K)$ for $K \geq 1$, the calculation requires that $\delta_1$ and $\delta_2$ be defined for all $K \geq 0$. (See the discussion leading to (13).) This example also helps emphasize that the T-convolution of privacy profiles corresponds to the composition of tradeoff functions and not to the composition of mechanisms. The composition of mechanisms in this setting leaves a bit unchanged with probability $p = p_1p_2 + (1-p_1)(1-p_2)$, and the resulting privacy profile has the form in (13) with this $p$.

We can informally think of $\delta_{P,R}$ as a measure of "distance" between probability measures $P$ and $R$. Measures that are "closer" together in this sense are harder to distinguish and thus provide better privacy guarantees. From this perspective, the next result proves a kind of triangle inequality, comparing the distance between $P$ and $R$ to their respective distances from a third measure $Q$. In the usual triangle inequality we get equality when two sides are perfectly aligned; an analogous condition yields equality in the current setting.

**Proposition 4.** *Suppose $P \ll\gg Q \ll\gg R$. Then (i)*

$$\delta_{Q,R} \bullet \delta_{P,Q} \geq \delta_{P,R}, \tag{26}$$

*and (ii) equality holds in (26) if $dQ/dR = \rho(dP/dQ)$ for some increasing function $\rho : \mathbb{R}_+ \to \mathbb{R}_+$.*

If we convert (26) to tradeoff functions, we get $T_{Q,R} \circ T_{P,Q} \geq T_{P,R}$. Lemma 7 in Dong et al. (2022) is similar but does not include the equality in part (ii) of the proposition.

**Example:** (Gaussian mechanism) Consider the mechanism that returns $X = y + Z$, $Z \sim N(0,1)$, when the true value is $y$. Let $y_p$, $y_q$, and $y_r$ be the true values for $P$, $Q$, and $R$. Then,

$$\frac{dQ}{dR} = e^{(y_q-y_r)X - (y_q^2 - y_r^2)/2}, \quad \frac{dP}{dQ} = e^{(y_p-y_q)X - (y_p^2 - y_q^2)/2},$$

so $dQ/dR$ is an increasing function of $dP/dQ$ if $y_r < y_q < y_p$ or $y_r > y_q > y_p$. In either of these cases, we get equality in (26). Theorem 3 of Dong et al. (2022) is a related result for tradeoff functions.

**Example:** (Beta mechanism). The equality attained by the Gaussian mechanism is a special case of a more general property for log-concave distributions, to which we return in Section 4.4. To illustrate that Proposition 4(ii) does not rely on this property we consider a mechanism that returns $X = U^{1/y}$ when the true value is $y > 0$, with $U$ uniform on $(0,1)$, which gives the response a beta distribution. For this mechanism,

$$\frac{dQ}{dR} = \frac{y_q}{y_r} X^{y_q - y_r}, \quad \frac{dP}{dQ} = \frac{y_p}{y_q} X^{y_p - y_q}.$$

Here, too, $dQ/dR$ is an increasing function of $dP/dQ$ if $y_r < y_q < y_p$ or $y_r > y_q > y_p$, even though the density of the response $X = U^{1/y}$ can be either log-concave ($y \geq 1$) or log-convex ($0 < y \leq 1$).

**Remark 1.** For part (ii) of Proposition 4, we could alternatively require that $dQ/dR$ and $dP/dQ$ be increasing functions of a common random variable. This condition implies (through Theorem 2 of Dhaene et al. (2002)) that $(dR/dQ, dP/dQ)$ have the joint distribution of $(G_{Q,R}^{-1}(U), G_{Q,P}^{-1}(1-U))$, which is the key property used in the proof. In either formulation, the condition is very strong; it assumes, in effect, that the same realization of the randomness in a mechanism is used for two separate queries. This will lead to a worst-case bound for group privacy in the next section.

### 4.3 Group Privacy

The symmetric neighboring relation $x \simeq x'$ on a collection of databases $\mathcal{X}$ usually means that the databases $x$ and $x'$ differ in at most one entry. Group privacy extends this notion to databases that may differ in up to $m$ entries. More precisely, write $x \simeq_m x'$ if $x \simeq x_1$, $x_1 \simeq x_2$, ..., $x_{m-1} \simeq x'$, for some databases $x_1, \dots, x_{m-1} \in \mathcal{X}$. Group privacy is concerned with how privacy guarantees for $\simeq$ extend to $\simeq_m$.

Let $M$ be a mechanism on $\mathcal{X}$ with privacy profile $\delta$; i.e., $\delta_{M(x),M(x')}(K) \leq \delta(K)$, for all $x \simeq x'$. As we take $\simeq$ to be symmetric, it is natural to assume that $\delta$ is symmetric. Write $\delta^{\bullet m}$ for the $m$-fold T-convolution of $\delta$ with itself: $\delta^{\bullet 1} = \delta$, and $\delta^{\bullet m} = \delta^{\bullet (m-1)} \bullet \delta$, for $m = 2, 3, \dots$. It follows immediately from Lemma 3 that symmetry of $\delta$ implies symmetry of $\delta^{\bullet m}$. The next result shows that the privacy profile $\delta$ extends to the group privacy profile $\delta^{\bullet m}$.

**Proposition 5.** *If $\delta_{M(x),M(x')}(K) \leq \delta(K)$, for all $x \simeq x'$, then $\delta_{M(x),M(x')}(K) \leq \delta^{\bullet m}(K)$, for all $x \simeq_m x'$.*

*Proof.* First suppose $m = 2$, and suppose $x_0 \simeq x_1 \simeq x_2$. Write $P_i$ for the distribution of $M(x_i)$. Then $\delta_{P_i,P_{i+1}} \leq \delta$, $i = 0, 1$. But then Proposition 4 implies that $\delta_{P_0,P_2} \leq \delta_{P_1,P_2} \bullet \delta_{P_0,P_1} \leq \delta \bullet \delta$. Because this holds for any $x_0 \simeq_2 x_2$, the proposition holds for $m = 2$. The extension to $m = 3, 4, \dots$ follows by induction. $\square$

As a special case of Proposition 5, we recover the bound of Vadhan (2017), Lemma 7.2.2, rephrased in terms of privacy profiles: if $M$ has privacy profile $\delta(K)$ with respect to $\simeq$, then it has privacy profile

$$\delta_m(K) = \frac{K - 1}{K^{1/m} - 1} \delta(K^{1/m}) \tag{27}$$

with respect to $\simeq_m$. In light of Proposition 5, it suffices to show that $\delta^{\bullet m} \leq \delta_m$. This bound holds with equality at $m = 1$. To proceed by induction, suppose the bound holds at $m$. Then

$$
\begin{aligned}
\delta^{\bullet (m+1)}(K^{m+1}) &= \delta \bullet \delta^{\bullet m}(K^{m+1}) \\
&\leq \delta \bullet \delta_m(K^{m+1}) \\
&\leq \delta(K) + K\delta_m(K^m), \quad \text{taking } \eta = K \text{ in (23)}, \\
&\leq \delta(K) + K\frac{K^m - 1}{K - 1}\delta(K), \quad \text{applying (27)}, \\
&= \frac{K^{m+1} - 1}{K - 1}\delta(K).
\end{aligned}
$$

Replacing $K$ with $K^{1/(m+1)}$ now shows that (27) holds at $m + 1$.

As discussed in Balle et al. (2020), the group privacy bound in (27) can often be improved in specific cases. However, we can use Proposition 4 to show that for any privacy profile $\delta$ there is a mechanism under which the bound in Proposition 5 becomes tight. More precisely, start with a probability measure $P_0$ and a random variable $L$ having cdf $1 + \delta'(\cdot)$ under $P_0$. Suppose $\delta$ is symmetric and regular; in particular, $L > 0$ and $\mathbb{E}_{P_0}[L] = 1$. Suppose further that $c_r = \mathbb{E}_{P_0}[L^r] < \infty$, $r = 2, \dots, m$, and let $c_1 = 1$. Define $P_i$, $i = 1, \dots, m$, by setting $dP_i/dP_0 = L^i/c_i$, $i = 1, \dots, m$. Interpret $P_i$ as the distribution of the response of a mechanism applied to a database $i$, and suppose consecutively indexed databases are neighbors. Then the group privacy profile for the mechanism is $\delta_{P_0,P_m}$, which, by repeated application of Proposition 4(ii) is equal to $\delta^{\bullet m}$. This worst-case bound is attained when all the likelihood ratios $dP_i/dP_0$ are increasing functions of the same random variable, in this case $L$.

Whereas Proposition 4 and the inequality $\delta^{\bullet m} \leq \delta_m$ provide upper bounds on T-convolutions, we also have the following simple lower bound. A corresponding result is provided by Tehranchi (2020) by a different argument.

**Lemma 5.** *For any privacy profiles $\delta_1, \delta_2$, we have $\delta_1 \bullet \delta_2 \geq \max\{\delta_1, \delta_2\}$.*

*Proof.* Let $\delta_0(K) = (1 - K)^+$. As $\delta_2(K) \geq \delta_0$, we have $\delta_1 \bullet \delta_2(K) \geq \delta_1 \bullet \delta_0(K) = \inf_{\eta \geq 0}\{\delta_1(\eta) + (\eta - K)^+\}$. We know that $\delta_1(\cdot)$ is decreasing with $\delta_1'(\eta) \geq -1$, for all $\eta$, whereas $(\eta - K)^+$ is increasing with slope 1 for

$\eta > K$, so the infimum is attained at $\eta = K$, where it equals $\delta_1(K)$. Thus, $\delta_1 \bullet \delta_2 \geq \delta_1$. A similar argument shows that $\delta_1 \bullet \delta_2 \geq \delta_2$. $\qquad\square$

Lemma 5 confirms that the group privacy profile bound $\delta^{\bullet m}$ is always at least as large as the privacy profile for neighboring databases, as it must be.

### 4.4  Additive Noise and Log-Concavity

We now restrict attention to mechanisms that return $Y = y + Z$ when the true query response for a database is $y \in \mathbb{R}$. (We can think of the query value $y$ as normalized relative to a reference value of zero and thus refer to $y$ as a sensitivity parameter, as in our discussion of the Gaussian mechanism. The results that follow depend only on differences in query values.) For the additive noise $Z$ we assume

$$Z \text{ has a density } f \text{ supported on all of } \mathbb{R}. \tag{28}$$

This condition could be weakened, but the assumption of full support ensures that responses for different values of $y$ are mutually absolutely continuous. Let $P_y$ denote the distribution of the response $Y$ when the true value is $y$; this response has density $f(\cdot - y)$. Let

$$\delta(K, y) = \mathbb{E}_{P_0}\left[\left(\frac{dP_y}{dP_0} - K\right)^+\right] = \mathbb{E}_{P_0}\left[\left(\frac{f(Y-y)}{f(Y)} - K\right)^+\right].$$

Then also, for any $x \in \mathbb{R}$,

$$\mathbb{E}_{P_x}\left[\left(\frac{dP_{x+y}}{dP_x} - K\right)^+\right] = \mathbb{E}_{P_x}\left[\left(\frac{f(Y-y-x)}{f(Y-x)} - K\right)^+\right] = \delta(K, y), \tag{29}$$

by a change of variables. In other words, the privacy profile $\delta_{P_y, P_{y'}}$ depends on the true values $y$ and $y'$ only through their difference $y - y'$. The following result is analogous to Theorem 1.1 of Tehranchi (2020).

**Proposition 6.** *For an additive noise mechanism satisfying (28), for any $K_1, K_2 > 0$ and any $y_1, y_2 \in \mathbb{R}$,*

$$\delta(K_1 K_2, y_1 + y_2) \leq \delta(K_1, y_1) + K_1 \delta(K_2, y_2). \tag{30}$$

*Proof.* Using (29) and then (26), we get, for any $K, \eta > 0$,

$$\delta(K, y_1 + y_2) = \delta_{P_0, P_{y_1+y_2}}(K) \leq \delta_{P_{y_2}, P_{y_1+y_2}} \bullet \delta_{P_0, P_{y_2}}(K) \leq \delta_{P_{y_2}, P_{y_1+y_2}}(K) + \eta \delta_{P_0, P_{y_2}}(K/\eta). \tag{31}$$

Taking $K = K_1 K_2$ and $\eta = K_1$ yields (30). $\qquad\square$

**Proposition 7.** *If $\log f$ is concave, then, for any $y_1, y_2 > 0$, and any $K \geq 0$,*

$$\delta(K, y_1 + y_2) = [\delta(\cdot, y_1) \bullet \delta(\cdot, y_2)](K). \tag{32}$$

*Proof.* Using (29), we have

$$\delta(K, y_1) = \mathbb{E}_{P_{y_2}}\left[\left(\frac{f(Y - y_1 - y_2)}{f(Y - y_2)} - K\right)^+\right], \quad \delta(K, y_2) = \mathbb{E}_{P_0}\left[\left(\frac{f(Y - y_2)}{f(Y)} - K\right)^+\right].$$

For log-concave $f$, a likelihood ratio of the form $f(Y - y - x)/f(Y - x)$, with $y > 0$, is an increasing function of $Y$ (see, e.g., Lehmann (1986), p.509). Thus, part (ii) of Proposition 4 applies (see Remark 1), so the first inequality in (31) becomes an equality. $\qquad\square$

Proposition 7 extends the identity (24) from the Gaussian mechanism to log-concave additive noise, showing that the T-convolution becomes additive in the sensitivity parameter $y$ in this setting. A related result for tradeoff functions appears in Proposition 9 of Dong et al. (2022), and a corresponding result for option prices in Theorem 3.8 of Tehranchi (2020).

Following Awan & Dong (2022), call a family of tradeoff functions $\{T_t, t \in [0, \infty)\}$ *infinitely divisible* if it satisfies the following properties:

(a′). $T_t \circ T_s = T_{t+s}$, for all $s, t \geq 0$;

(b′). for all $s > 0$, $T_s$ is nontrivial, meaning that $0 < T_s(x) < x$, for some $x \in (0, 1)$;

(c′). as $s \downarrow 0$, $T_s(x) \to T_0(x) = x$, for all $x \in [0, 1]$.

(We have added the lower bound in (b′) to the conditions in Awan & Dong (2022) to exclude $T_s \equiv 0$.) Awan & Dong (2022) call a tradeoff function infinitely divisible if it is an element of such a family. We can similarly introduce the following definition for privacy profiles, replacing tradeoff composition in (a′) with T-convolution in (a).

**Definition 7.** *A family of privacy profiles $\{\delta_t, t \geq 0\}$ is* infinitely divisible *if it satisfies the following properties:*

*(a)* $\delta_t \bullet \delta_s = \delta_{t+s}$, *for all $s, t \geq 0$;*

*(b)* *for all $s > 0$, $\delta_s$ satisfies $0 < \delta_s(1) < 1 + \delta_s'(1) < 1$;*

*(c)* *as $s \downarrow 0$, $\delta_s(K) \to \delta_0(K) = (1 - K)^+$, for all $K \in [0, \infty]$.*

In (b), convexity implies (see Appendix A) $\delta_s(1) \leq 1 + \delta_s'(1)$ because $\delta_s(0) = 1$, so the main requirement is that this inequality be strict. In (c), as before, $\delta(\infty) = \lim_{K \to \infty} \delta(K) = \inf_{K \geq 0} \delta(K)$. For the following, recall that we defined $T_\delta(\alpha) = 1 + \delta^*(\alpha - 1)$ in (19).

**Lemma 6.** *The privacy profiles $\{\delta_t, t \in [0, \infty)\}$ are infinitely divisible if and only if the tradeoff functions $\{T_{\delta_t}, t \in [0, \infty)\}$ are infinitely divisible.*

Theorem 3.3 of Awan & Dong (2022) characterizes the tradeoff functions of log-concave mechanisms as elements of infinitely divisible families. (A precise statement of their result relies on the notion of a canonical noise distribution.) Through the equivalence in Lemma 6, their result extends to a characterization through privacy profiles.

## 5 Convexity Properties of Privacy Profiles for Log-Concave Additive Noise

This section establishes two properties of the shape of privacy profiles for log-concave additive noise. Theorem 2 shows that they are sigmoidal in the sensitivity parameter $y$, meaning that they are convex for small values of $y$ and concave for larger values. Theorem 3 establishes a *relative* convexity property, showing that, in a precise sense, $\delta(K, y)$ is more convex in $y$ at larger values of $K$. Convexity in $y$ reflects accelerating privacy loss.

### 5.1 Sigmoidal Property

Dong et al. (2021) show that the tradeoff functions for mechanisms of the form $q(x) + \sigma Z$ (where $q(x)$ is the true query value for database $x$) are pointwise increasing (more private) in $\sigma$ if $Z$ is log-concave. By applying Lemma 5 to (32), we get a corresponding result for privacy profiles, which is a special case of Lemma 18 of Vinterbo (2022).

**Proposition 8.** *For log-concave additive noise, $\delta(K, y)$ is increasing in $y > 0$, for all $K \geq 0$.*

Increasing $y$ corresponds to increasing the difference in query values between databases or decreasing the scaling of the additive noise, both of which reduce privacy, as reflected in the reduction in $\delta$.

We turn next to second-order properties of $\delta(K, y)$ in $y$. (Convexity in $K$ is automatic.) Convexity in $y$ indicates an accelerating loss of privacy as $y$ increases, whereas concavity indicates a slowing rate of loss. For the rest of this section, we impose the following conditions on the additive noise density $f$.

For all $z \in \mathbb{R}$, $f(z) = e^{-\psi(z)}$, where $\psi$ is even, $\psi'(z) > 0$, for all $z > 0$, and $\psi''(z) > 0$, for all $z \in \mathbb{R}$.  (33)

These conditions are stronger than strictly necessary, but they simplify several arguments. The condition $\psi'' > 0$ makes $f$ (strictly) log-concave. A log-concave density is unimodal (see, e.g., Lehmann (1986), p.509); the assumption that $\psi$ is even and strictly increasing for $z < 0$ puts the mode of $f$ at zero. The assumption that $\psi$ (and therefore $f$) is even makes the additive noise symmetric, which is typical of the examples commonly considered.

Define the likelihood ratio

$$\Lambda(z, y) = f(z - y)/f(z), \tag{34}$$

and the inverse function

$$t(K, y) = \sup\{z : \Lambda(z, y) \leq K\}. \tag{35}$$

Strict convexity of $\psi$ on $\mathbb{R}_{++}$ implies that $\Lambda(\cdot, y)$ is strictly increasing, for all $y > 0$; it increases toward

$$K_{\max}(y) = \lim_{z \uparrow \infty} \Lambda(z, y),$$

which may be infinite. It follows from (33) that $f(z - y)$ is increasing in $y$, for $z > y$, and therefore that $\Lambda(z, y)$ is increasing in $y$, for all $z > y$, and therefore that $K_{\max}(y)$ is increasing in $y$. Because $f$ is even,

$$\Lambda(y/2, y) = f(-y/2)/f(y/2) = 1.$$

Thus, for any $y > 0$, $\Lambda(\cdot, y)$ maps $[y/2, \infty)$ onto $[1, K_{\max}(y))$, and $\Lambda(\cdot, y)$ has a well-defined inverse $t(\cdot, y) : [1, K_{\max}(y)) \to [y/2, \infty)$,

$$\Lambda(t(K, y), y) = K, \quad 1 \leq K < K_{\max}(y). \tag{36}$$

For $K > K_{\max}(y)$, take $t(K, y) = \infty$, as in (35). Equation (36) may be valid for $K < 1$, but we limit our discussion to $K \geq 1$, which corresponds to $\epsilon \geq 0$ in the usual parameterization of $(\epsilon, \delta)$ differential privacy.

The inverse likelihood ratio $t(K, y)$ leads to an explicit expression for the privacy profile for the additive noise mechanism with noise density $f$. As in Lemma 1 of Vinterbo (2022), we have

$$\delta(K, y) = F(y - t(K, y)) - KF(-t(K, y)), \tag{37}$$

where $F$ is the cdf for the density $f$. This in turn leads to an expression for the derivative of the privacy profile with respect to the sensitivity parameter $y$. We write $\partial_y$ and $\partial_K$ for partial derivatives with respect to $y$ and $K$.

**Lemma 7.** *We have*

$$\partial_y \delta(K, y) = f(t(K, y) - y) = Kf(t(K, y)). \tag{38}$$

We can now characterize the regions of convexity and concavity of the privacy profile with respect to the sensitivity parameter $y$; these are the intervals on which $\partial_y \delta(K, \cdot)$ are increasing and decreasing, respectively.

**Theorem 2.** *For $K > 1$, let $y_*$ be the unique solution to $K = f(0)/f(y)$. Then $\delta(K, \cdot)$ is convex on $(0, y_*)$ and concave on $(y^*, \infty)$.*

**Example:** (Gaussian mechanism.) The standard Gaussian density satisfies (33). In this case, we have $K_{\max}(y) = \infty$, $t(K, y) = \log K/y + y/2$, and the equation $K = f(0)/f(y)$ is solved by $y^* = \sqrt{2 \log K}$. As a function of $y$, the privacy profile (11) is convex on $(0, \sqrt{2 \log K})$ and concave on $(\sqrt{2 \log K}, \infty)$. The left panel of Figure 2 plots $\delta(K, y)$, for $K = 2$. The circled point is the inflection point $y^*$, where the profile switches from convex to concave.

Recalling (32), we may express Theorem 2 as saying that the difference in profiles

$$[\delta(\cdot, y_1) \bullet \delta(\cdot, y_2)](K) - \delta(K, y_1)$$

is increasing in $y_2$, for $0 < y_1 < y_1 + y_2 < y^*$, and decreasing in $y_2$, for $y^* < y_1 < y_1 + y_2$.

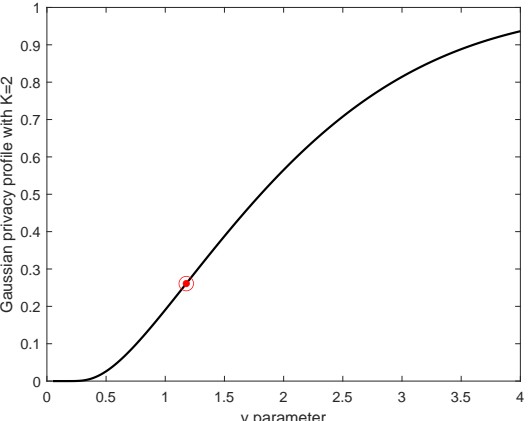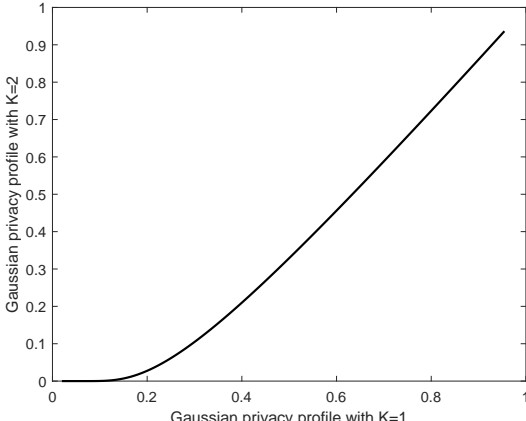

Figure 2: Left: The Gaussian privacy profile $\delta(2, y)$ is convex for small $y$ and concave for large $y$, the transition occurring at the circled point $y^*$. Right: $\delta(2, y)$ is convex when plotted against $\delta(1, y)$.

## 5.2 Relative Convexity

We have seen that log-concave privacy profiles $\delta(K, y)$ are sigmoidal in $y$ and neither globally convex nor globally concave. We will now show that under a widely applicable additional condition, log-concave privacy profiles enjoy a *relative* convexity property: for $1 \leq K_1 < K_2$, this means that $\delta(K_2, \cdot) \circ \delta(K_1, \cdot)^{-1}$ is convex, where $\delta(K, \cdot)^{-1}$ denotes the inverse with respect to the second argument. (Recall from Proposition 8 that $\delta(K, y)$ is increasing in $y$.) When this holds, we say that $\delta(K_2, \cdot)$ is *convex with respect to $\delta(K_1, \cdot)$*. This property says that the higher-$K$ profile is "more convex" than the lower-$K$ profile. Graphically, it means that a plot of the curve $y \rightarrow (\delta(K_1, y), \delta(K_2, y))$ is convex, even though the plot of $(y, \delta(K, y))$ is not. This is illustrated in the right panel of Figure 2, which shows that $\delta(2, y)$ becomes convex when plotted against $\delta(1, y)$.

Relative convexity leads to bounds on privacy loss, which we discuss after the following result. As in (33), $\psi = -\log f$.

**Theorem 3.** *Suppose*

$$\psi''(z)/[\psi'(z)]^2 \quad \text{is decreasing in } z > 0. \tag{39}$$

*Then for $1 \leq K_1 < K_2$, $\delta(K_2, \cdot)$ is convex with respect to $\delta(K_1, \cdot)$, on $(y_0, \infty)$, for all $y_0 > 0$ with $K_{\max}(y_0) \geq K_2$.*

If $K_{\max}(y) = \infty$ for all $y > 0$, then the conclusion holds for any $y_0 > 0$.

We will prove the theorem by proving that condition (40) in the following lemma holds. The lemma specializes Theorem 1 of Cargo (1965) to our setting. Recall that $\delta(K, y)$ is increasing in $y$.

**Lemma 8.** *If*

$$\frac{\partial_y \delta(K_2, y)}{\partial_y \delta(K_1, y)} \quad \text{is increasing in } y, \ y > y_0, \tag{40}$$

*then $\delta(K_2, \cdot)$ is convex with respect to $\delta(K_1, \cdot)$, on $(y_0, \infty)$.*

From (38), we get

$$\frac{\partial_y \delta(K_2, y)}{\partial_y \delta(K_1, y)} = \frac{K_2 f(t(K_2, y))}{K_1 f(t(K_1, y))} = \frac{f(t(K_2, y) - y)}{f(t(K_1, y) - y)}, \tag{41}$$

so it will suffice to show that the rightmost expression in (41) is increasing in $y$ for $K_2 > K_1$.

We can interpret relative convexity as saying that the privacy loss at $K_2$ accelerates relative to the privacy loss at $K_1$, as $y$ increases. In particular, with $1 \leq K_1 < K_2$, and $0 < y_1 < y_2$, Proposition 2.6 of Rajba

| Distribution | $\psi(z)$ | on $\mathbb{R}_{++}$ | | |
|---|---|---|---|---|
| | | $\psi' > 0$ | $\psi'' > 0$ | $\psi''/\psi'^2$ decreasing |
| Normal | $z^2/2$ | ✓ | ✓ | ✓ |
| Subbotin | $|z|^r$ | ✓ | $r > 1$ | $r > 1$ |
| Logistic | $z + 2\log(1 + e^{-z})$ | ✓ | ✓ | ✓ |
| Sym-Gamma | $|z| + \alpha - \alpha\log(|z| + \alpha)$ | $\alpha > 0$ | $\alpha > 0$ | $\alpha > 0$ |

Table 1: Examples of log-concave noise distributions satisfying (39)

(2014) and (41) yield

$$\delta(K_2, y_2) - \delta(K_2, y_1) \geq \frac{K_2 f(t(K_2, y_1))}{K_1 f(t(K_1, y_1))}(\delta(K_1, y_2) - \delta(K_1, y_1)).$$

Recall than an increase in the sensitivity parameter $y$ entails a loss of privacy, which is reflected in an increase in the privacy profile. The inequality bounds the privacy loss at a higher value of $K$ (a higher value of $\epsilon$ with $K = e^\epsilon$) relative to the privacy loss at a lower value of $K$.

In the case of the Gaussian additive noise, at $K = 1$ (11) simplifies to $\delta(1, y) = 2\Phi(y/2) - 1$. If we apply Theorem 3 with $K_1 = 1$ and $K_2 = K > 1$, we get the following consequence. Suppose the sensitivity parameters $0 < y_1 < y_2 < y_3$ satisfy

$$\Phi(y_2/2) - \Phi(y_1/2) \leq \Phi(y_3/2) - \Phi(y_2/2); \tag{42}$$

then

$$\delta(K, y_2) - \delta(K, y_1) \leq \delta(K, y_3) - \delta(K, y_2). \tag{43}$$

If $\delta(K, \cdot)$ were convex, then (43) would hold whenever the increment $y_3 - y_2$ exceeds the increment $y_2 - y_1$. Theorem 3 tells us that the required spacing of the increments is instead determined by (42). For sensitivity parameters satisfying (42), the privacy loss increments satisfy (43). A version of (42)–(43) appears in Glasserman & Pirjol (2023) in the context of option pricing.

Table 1 shows examples of log-concave noise distributions and the parameter ranges in which they satisfy the conditions of this section. The first three are also studied in Vinterbo (2022). The last example is a symmetrized gamma distribution. The gamma density proportional to $z^\alpha e^{-z}$ on $\mathbb{R}_+$ is log-concave for $\alpha > 0$, in which case $\alpha$ is also the mode. The function $(z + \alpha)^\alpha e^{-(z+\alpha)}$ is therefore decreasing for $z > 0$. Combining it with its mirror image for $z < 0$ yields a symmetric log-concave function on $\mathbb{R}$ that can be normalized to a probability density. We omit some normalization constants in Table 1 because they do not affect the conditions on $\psi$.

The Subbotin distribution with $r = 1$ is the Laplace distribution. By requiring $r > 1$ in Table 1, we are requiring strict log-concavity and excluding this case; for the Laplace distribution, $\psi'' = 0$. The privacy profile for the Laplace distribution takes the form (see Theorem 3 of Balle et al. (2020)) $\delta(K, y) = (1 - \sqrt{K}e^{-y/2})^+$. If we take $y > \log K_2 > \log K_1 \geq 0$ and check for relative convexity we get $\delta(K_2, \cdot)^{-1} \circ \delta(K_1, y) = y + \log(K_2/K_1)$, which is linear in $y$. The notion of relative convexity thus degenerates in this setting — neither function $\delta(K_i, \cdot)$, $i = 1, 2$, is more convex than the other.

## 6 Concluding Remarks

This paper contributes to the characterization of privacy mechanisms by establishing connections between privacy profiles and tradeoff functions, two frameworks for describing differential privacy. Our main result shows that the composition of tradeoff functions corresponds to the T-convolution of privacy profiles. These operations are relevant because the composition of tradeoff functions characterizes group privacy and log-concave mechanisms, which include the Gaussian and Laplace mechanisms. This paper also characterizes *regular* privacy profiles, which are privacy profiles for mutually absolutely continuous probability measures. Regularity is determined by two boundary conditions, which also translate to simple conditions on tradeoff functions. We also derive new convexity properties for log-concave mechanisms, which provide new bounds on their privacy profiles. Some of these results exploit analogies between privacy profiles and option pricing.

## A    Background Results

This appendix provides some background technical results, primarily concerning properties of convex functions. These properties are used throughout the paper because privacy profiles $\delta(\cdot)$ are convex functions on $[0, \infty)$, and tradeoff functions $T(\cdot)$ are convex functions on $[0, 1]$.

Let $I \subset \mathbb{R}$ be an open interval and suppose $f : I \to \mathbb{R}$ is convex. Denote by $f'_+$ and $f'_-$ the right- and left-derivatives of $f$. The following summarizes results from Theorem 2.1.2 and Corollary 2.1.3 of Borwein & Vanderwerff (2010).

(i)  $f'_+(x)$ and $f'_-(x)$ exist and are finite at every $x \in I$;

(ii)  $f'_+$ and $f'_-$ are increasing functions on $I$;

(iii)  $f'_+(x) \leq f'_-(y) \leq f'_+(y)$, for $x < y$ with $x, y \in I$;

(iv)  if $f'_-(x) \leq \lambda \leq f'_+(x_0)$, then $f(y) \geq f(x) + \lambda(y - x)$, for all $y \in I$.

(v)  for any $x, y \in I$,

$$f(y) - f(x) = \int_x^y f'_-(u)\, du = \int_x^y f'_+(u)\, du. \tag{44}$$

The closed interval $[f'_-(x), f'_+(x)]$ used in (iv) is the subdifferential of $f$ at $x$, denoted by $\partial f(x)$.

To lighten notation, throughout this paper we write $\delta'$ for the right-derivative of $\delta$. As an application of (iv), with $f = \delta$, $f'_+ = \delta'$, $x = 1$, $y = 0$, we have $\delta(0) \geq \delta(1) - \delta'(1)$, and thus $\delta(1) \leq 1 + \delta'(1)$, a property used following Definition 7. Similarly, $\delta(x) - \delta(0) \leq x\delta'(x)$, a property used in the proof of Proposition 2.

Part (v) does not appear explicitly in Borwein & Vanderwerff (2010), but their Theorem 2.1.2(f) implies (v).

As in (17), for a function $g : \mathbb{R} \to \mathbb{R}$, the convex conjugate $g^* : \mathbb{R} \to \mathbb{R}$ is defined by setting

$$g^*(y) = \sup_{x \in \mathbb{R}} \{yx - g(x)\}.$$

If the domain of $g$ is a subinterval of $\mathbb{R}$, the supremum is taken over that subinterval. In particular,

$$\delta^*(y) = \sup_{K \geq 0} \{yK - \delta(K)\}, \quad T^*(y) = \sup_{\alpha \in [0,1]} \{y\alpha - T(\alpha)\}.$$

The conjugate of the conjugate is denoted by $g^{**}$. Proposition 4.4.2 of Borwein & Vanderwerff (2010) gives general conditions under which $g^{**} = g$. The following sufficient condition, which follows from their result, is adequate for our purposes because it applies to both privacy profiles and tradeoff functions:

(vi)  if $g$ is finite, convex, and continuous on an interval $[a, b]$ or $[a, \infty)$, then $g^{**} = g$ on that interval.

Thus, $\delta^{**} = \delta$ and $T^{**} = T$.

We also use the following consequence of Proposition 4.4.5 of Borwein & Vanderwerff (2010), which relates the subdifferentials (defined above) of $g$ and $g^*$. This property is used in the proof of Proposition 3.

(vii)  if $g$ is finite, convex, and continuous on an interval $[a, b]$ or $[a, \infty)$, then for any $x$ in that interval, $u \in \partial g(x)$ if and only if $x \in \partial g^*(u)$.

Thr proof of Lemma 6 requires moving back and forth between convergence of a sequence of privacy profiles and convergence of a sequence of tradeoff functions. This in turn requires interchanging limits and convex conjugation. Theorem 1.3 of Komuro (1989) gives conditions permitting this interchange. It implies the following sufficient condition in particular:

(viii) Suppose the functions $g$ and $g_n$, $n \geq 1$, are finite and convex on an interval $I$ and that $g_n(x) \to g(x)$ for all $x \in I$. If $I = [0, 1]$, then $g_n^*(y) \to g^*(y)$, for all $y \in \mathbb{R}$, and if $I = [0, \infty)$, then $g_n^*(y) \to g^*(y)$, for all $y < 0$.

The two cases for the interval $I$ apply to tradeoff functions and privacy profiles, respectively, allowing us to formulate the following result, based on the mappings in (18) and (19) between privacy profiles and tradeoff functions.

**Lemma 9.** *If $\delta_n(K) \to \delta(K)$, for all $K \geq 0$, then $T_{\delta_n}(\alpha) \to T_\delta(\alpha)$, for all $\alpha \in [0, 1)$. If $T_n(\alpha) \to T(\alpha)$, for all $\alpha \in [0, 1]$, then $\delta_{T_n}(K) \to \delta_T(K)$, for all $K \geq 0$.*

*Proof.* If $\delta_n(K) \to \delta(K)$, for all $K \geq 0$, then $\delta_n^*(y) \to \delta^*(y)$, for all $y < 0$, and then, through (19), $T_{\delta_n}(\alpha) \to T_\delta(\alpha)$, for all $\alpha < 1$. The second claim works similarly. $\square$

## B  Omitted Proofs from Section 3

*Proof of Lemma 1.* Convexity of $\delta$ implies that (the right-derivative) $\delta'$ is increasing (see Appendix A), so $G$ is increasing, with $G(0) = 1 + \delta'(0) = 0$. Because $\delta$ is decreasing, $\delta'(x) \leq 0$, for all $x \geq 0$, so $G(x) \leq 1$. A convex function is equal to the integral of its right-derivative (see (44)), so, for any $0 \leq K < K'$,

$$\delta(K) - \delta(K') = -\int_K^{K'} \delta'(x)\,dx = \int_K^{K'} [1 - G(x)]\,dx.$$

Letting $K' \to \infty$, we get (by regularity) $\delta(K') \to 0$ and therefore

$$\delta(K) = \int_K^\infty [1 - G(x)]\,dx.$$

In particular, $G(x) \to 1$ as $x \to \infty$. Thus, $G$ is a cumulative distribution function. If $L$ has distribution $G$ under $P$, then the first equality in (9) follows from integration by parts. $\square$

*Proof of Proposition 1.* For the "if" direction, let $\delta$ be a regular privacy profile. Let the random variable $L$ have distribution $G$ under some probability measure $P$, with $G = 1 + \delta'$, as in Lemma 1. Then $P(L = 0) = G(0) = 0$, and

$$\mathbb{E}_P[L] = \int_0^\infty [1 - G(x)]\,dx = \delta(0) = 1.$$

Thus, $L$ is positive and has unit expectation under $P$. We may therefore define a probability measure $Q$ by setting $Q(A) = \mathbb{E}_P[L\mathbf{1}_A]$, for measurable $A$. In other words, $dQ/dP = L$, and because $L$ is strictly positive and finite, $P \lll Q$. Moreover, by Lemma 1,

$$\mathbb{E}_P\left[\left(\frac{dQ}{dP} - K\right)^+\right] = \mathbb{E}_P[(L - K)^+] = \delta(K), \tag{45}$$

so $\delta$ is indeed the privacy profile for mutually absolutely continuous probability measures $P$ and $Q$.

To prove the "only if" claim, suppose $Q \ll P$. Then $\mathbb{E}_P[dQ/dP] = 1$, and the dominated convergence theorem implies

$$\lim_{K \to \infty} \delta(K) = \lim_{K \to \infty} \mathbb{E}_P\left[\left(\frac{dQ}{dP} - K\right)^+\right] = \mathbb{E}_P\left[\lim_{K \to \infty} \left(\frac{dQ}{dP} - K\right)^+\right] = 0,$$

confirming that $\delta(\infty) = 0$. From the representation

$$\mathbb{E}_P\left[\left(\frac{dQ}{dP} - K\right)^+\right] = \int_K^\infty P(dQ/dP > x)\,dx,$$

we get $\delta'(0) = -P(dQ/dP > 0)$. If $P \ll Q$, then $P(dQ/dP = 0) = 0$, so $\delta'(0) = -1$. $\square$

*Proof of Proposition 2.* (i) Using (15) and (2), we get

$$
\begin{aligned}
\hat{\delta}_{P,Q}(K) &= 1 - K + K\delta_{P,Q}(1/K) \\
&= 1 - K + \sup_{A}\{KQ(A) - P(A)\} \\
&= 1 - K + \sup_{A}\{K(1 - Q(A)) - (1 - P(A))\} \\
&= \sup_{A}\{P(A) - KQ(A)\} = \delta_{Q,P}(K).
\end{aligned}
$$

(ii) We will verify that $\hat{\delta}$ satisfies the properties in (8) in the order they appear there. By Boyd & Vandenberghe (2004), p.89, if $\delta(K)$ is convex then so is $K\delta(1/K)$. To show that $\hat{\delta}$ is decreasing, we need to show (taking right-derivatives)

$$
\hat{\delta}'(K) = -1 + \delta(1/K) - \delta'(1/K)/K \le 0, \tag{46}
$$

for $K > 0$. Convexity of $\delta$ implies (see Appendix A) that $\delta(x) - \delta(0) \le x\delta'(x)$. At $x = 1/K$, this yields $-\delta(0) + \delta(1/K) - \delta'(K)/K \le 0$, and recalling that $\delta(0) = 1$ yields (46). At $K = 0$, we have $\hat{\delta}(0) = 1$. Finally, we have

$$
\hat{\delta}(K) = 1 - K + K\delta(1/K) \ge 1 - K + K(1 - 1/K)^+ = 1 - K + (K - 1)^+ = (1 - K)^+.
$$

The fact that $\hat{\hat{\delta}} = \delta$ is immediate from (15). (iii) It remains to show that $\delta$ and $\hat{\delta}$ share the properties in Proposition 1. From $(\hat{\delta}(K) - \hat{\delta}(0))/K = -1 + \delta(1/K)$, we see that $\hat{\delta}'(0) = -1 \Leftrightarrow \delta(\infty) = 0$. Similarly,

$$
\delta'(0) = -1 \Leftrightarrow \hat{\delta}(\infty) = 0.
$$

Thus, if either profile is regular, then so is the other. $\qquad\square$

## C   Omitted Proofs from Section 4

The following result is stated as Lemma 20 of Zhu et al. (2022) as a consequence of Proposition 6 of Dong et al. (2022). For completeness, we provide a proof without the symmetry condition in Proposition 6 of Dong et al. (2022).

**Lemma 10.** *For a pair of probability measures $(P, Q)$, let $\delta$ be the privacy profile and let $T_o$ be the tradeoff function under the definition in (6). Then $\delta(K) = 1 + T_o^*(-K)$, for all $K \ge 0$.*

*Proof.* Recall the hypothesis testing interpretation of $T_o$, which says that $T_o(\alpha)$ is the smallest type II error probability among tests with type I error probability $\alpha$, with $P$ the distribution under the null hypothesis and $Q$ the distribution under the alternative. For any measurable set $A$, the test with rejection set $A$ has type I error probability $\alpha = P(A)$ and type II error probability $1 - Q(A) \ge T_o(\alpha)$. Thus, recalling (2),

$$
\delta(K) = \sup_{A}\{Q(A) - KP(A)\} \le \sup_{\alpha}\{1 - T_o(\alpha) - K\alpha\} = 1 + \sup_{\alpha}\{-K\alpha - T_o(\alpha)\} = 1 + T_o^*(-K). \tag{47}
$$

To prove the reverse inequality, we recall that the Neyman-Pearson lemma (as in Lehmann (1986), pp.74–75) says that for all $\alpha$ there exist constants $k > 0$, $c \in [0, 1]$, and a test

$$
\phi = \begin{cases} 1, & dQ/dP > k; \\ c, & dQ/dP = k; \\ 0, & dQ/dP < k, \end{cases} \tag{48}
$$

such that $\mathbb{E}_P[\phi] = \alpha$ and $T_o(\alpha) = 1 - \mathbb{E}_Q[\phi]$.

In (48), we allow the possibility that $dQ/dP = \infty$, but consider first the case $Q \ll P$. Note that $(x - K)^+ \ge u(x - K)$, for all $u \in [0, 1]$. As $\phi$ takes values in $[0, 1]$, we therefore have

$$
\begin{aligned}
\delta(K) = \mathbb{E}_P[(dQ/dP - K)^+] &\ge \mathbb{E}_P[(dQ/dP)\phi] - K\mathbb{E}_P[\phi] \\
&= \mathbb{E}_Q[\phi] - K\mathbb{E}_P[\phi] \\
&= 1 - T_o(\alpha) - K\alpha.
\end{aligned}
$$

As this holds for all $\alpha$, we have

$$\delta(K) \geq 1 + \sup_\alpha\{-K\alpha - T_o(\alpha)\} = 1 + T_o^*(-K). \tag{49}$$

Without the assumption that $Q \ll P$, we have the more general representation in (4), which yields $\delta(K) = \mathbb{E}_P[(dQ/dP - K)^+ \mathbf{1}_A] + Q(A^c)$, with $P(A) = 1$. Since $P(A^c) = 0$, we have $\phi = 1$ on $A^c$; an optimal test always rejects $P$ on $A^c$. Thus, $\mathbb{E}_P[(dQ/dP)\phi\mathbf{1}_A] + Q(A^c) = \mathbb{E}_Q[\phi]$, and $\mathbb{E}_P[\phi\mathbf{1}_A] = \mathbb{E}_P[\phi] = \alpha$, so

$$\delta(K) = \mathbb{E}_P[(dQ/dP - K)^+ \mathbf{1}_A] + Q(A^c) \geq \mathbb{E}_P[(dQ/dP)\phi\mathbf{1}_A] + Q(A^c) - K\mathbb{E}_P[\mathbf{1}_A\phi] = 1 - T_o(\alpha) - K\alpha,$$

and we again get (49). Combining (47) and (49) proves the lemma. $\qquad\square$

We briefly discuss how this result for an ordered pair of probability measures $(P, Q)$ relates to the symmetric result for mechansims in Proposition 6 of Dong et al. (2022). The argument in Lemma 10 can be reformulated to say that $(P, Q)$ satisfies $(\epsilon, \delta)$-DP (1) if and only if $T_o(\alpha) \geq 1 - \delta - K\alpha$, with $K = e^\epsilon$ (and $\delta(K)$ is then the smallest $\delta$ for which this holds). To make this a symmetric condition we would apply it as well to $T_o^{-1}$, which, as discussed in Dong et al. (2022), is the tradeoff function for the pair $(Q, P)$. But, taking inverses of both sides, the condition $T_o^{-1}(\alpha) \geq 1 - \delta - K\alpha$ is equivalent to $T_o(\alpha) \geq (1/K)(1 - \delta - \alpha)$. Recalling that tradeoff functions are nonnegative, we can write the combined conditions on $(P, Q)$ and $(Q, P)$ as

$$T_o(\alpha) \geq f_{\epsilon,\delta}(\alpha) = \max\{0, 1 - \delta - e^\epsilon\alpha, e^{-\epsilon}(1 - \delta - \alpha)\}. \tag{50}$$

This is the condition in Proposition 3 of Dong et al. (2022), based on extending the argument in Theorem 2.4 of Wasserman & Zhou (2010). Condition (50) is the symmetrized version of the condition $T_o(\alpha) \geq 1 - \delta - K\alpha$.

*Proof of Proposition 3.* From (20) we get $T_\delta(1) = 1 + \delta^*(0) = 1 + \sup_K\{-\delta(K)\} = 1 - \delta(\infty)$, so $T_\delta(1) = 1 \Leftrightarrow \delta(\infty) = 0$. The condition $\delta(K) \geq (1 - K)^+$ in (8) implies that $\delta'(0) \geq -1$, so for regularity is suffices to show that $\delta'(0) \leq -1$. Through (21), this is equivalent to $T^{*\prime}(0) \leq 0$, where $T^{*\prime}$ denotes the right-derivative of $T^*$. Thus, we have

$$\begin{aligned}
\delta'(0) = -1 \quad &\Leftrightarrow \quad T_\delta^{*\prime}(0) \leq 0 \\
&\Leftrightarrow \quad \nexists b > 0 \text{ such that } b \in \partial T_\delta^*(0) \\
&\Leftrightarrow \quad \nexists b > 0 \text{ such that } 0 \in \partial T_\delta(b) \\
&\Leftrightarrow \quad \nexists b > 0 \text{ such that } T_\delta(\alpha) \geq T_\delta(b), \text{ for all } \alpha \in [0, 1] \\
&\Leftrightarrow \quad T_\delta(b) > T_\delta(0) = 0, \text{ for all } b \in (0, 1].
\end{aligned}$$

The second equivalence holds because the right-derivative is the largest element of the subdifferential $\partial T_\delta^*(0)$, as discussed in Appendix A. The third step uses the inverse relation between the subdifferentials of convex conjugates noted in item (vii) of Appendix A. For the fourth step, $v \in \partial T_\delta(b)$ means that $T_\delta(\alpha) \geq T_\delta(b) + v(\alpha - b)$, for all $\alpha$, (by item (iv) of Appendix A) and this becomes $T_\delta(\alpha) \geq T_\delta(b)$ when $v = 0$. The last step holds because $T_\delta$ attains its minimum value zero at zero. $\qquad\square$

*Proof of Lemma 3.* We have

$$\widehat{\delta_1 \bullet \delta_2}(K) = 1 - K + K(\delta_1 \bullet \delta_2)(1/K) = 1 - K + K\inf_{\eta \geq 0}\{\delta_1(\eta) + \eta\delta_2(1/\eta K)\},$$

and also

$$\begin{aligned}
\hat{\delta}_2 \bullet \hat{\delta}_1(K) &= \inf_{\eta' \geq 0}\{\hat{\delta}_2(\eta') + \eta'\hat{\delta}_1(K/\eta')\} \\
&= \inf_{\eta' \geq 0}\{1 - \eta' + \eta'\delta_2(1/\eta') + \eta'[1 - K/\eta' + (K/\eta')\delta_1(\eta'/K)]\} \\
&= 1 - K + \inf_{\eta' \geq 0}\{\eta'\delta_2(1/\eta') + K\delta_1(\eta'/K)\} \\
&= 1 - K + K\inf_{\eta \geq 0}\{\eta\delta_2(1/\eta K) + \delta_1(\eta)\},
\end{aligned}$$

where the last step follows from setting $\eta = \eta'/K$. $\qquad\square$

*Proof of Lemma 4.* (i) We verify that $\delta_1 \bullet \delta_2$ satisfies the properties in (8). For each $\eta$, $\delta_1(\eta) + \eta\delta_2(K/\eta)$ is decreasing in $K$ so the same holds for the infimum over $\eta$. For any $\eta \geq 0$,

$$\delta_1(\eta) + \eta\delta_2(K/\eta) \geq (1-\eta)^+ + \eta(1-K/\eta)^+ \geq (1-\eta)^+ + (\eta-K)^+ \geq (1-K)^+,$$

with $0 \cdot (1 - K/0)^+ = 0$, so $\delta_1 \bullet \delta_2(K) \geq (1-K)^+$, and $\delta_1 \bullet \delta_2(0) = 1$.

It remains to show convexity. By Boyd & Vandenberghe (2004), p.89, the function $(\eta, K) \mapsto \eta\delta_2(K/\eta)$ inherits convexty from $\delta_2$. Then $(\eta, K) \mapsto \delta_1(\eta) + \delta_2(K/\eta)$ is also convex. By Boyd & Vandenberghe (2004), pp.87–88, taking the infimum over $\eta$ preserves convexity, so $\delta_1 \bullet \delta_2$ is convex.

(ii) Now suppose $\delta_1$ and $\delta_2$ are regular. We verify that $\delta_1 \bullet \delta_2$ satisfies the two properties in Definition 5. For any $\eta > 0$,
$$0 \leq \delta_1 \bullet \delta_2(K) \leq \delta_1(\eta) + \eta\delta_2(K/\eta).$$
Because $\delta_2$ is regular, $\delta_2(K/\eta) \to 0$ as $K \to \infty$, so

$$0 \leq \delta_1 \bullet \delta_2(\infty) \leq \delta_1(\eta),$$

and since this holds for all $\eta > 0$, and $\delta_1$ is regular, we conclude that that $\delta_1 \bullet \delta_2(\infty) = 0$.

To complete the proof, we need to show that $(\delta_1 \bullet \delta_2)'(0) = -1$. In light of (16), it suffices to show $\widehat{\delta_1 \bullet \delta_2}(\infty) = 0$. By Lemma 3 this condition is equivalent to $\hat{\delta}_2 \bullet \hat{\delta}_1(\infty) = 0$. By Proposition 2(iii), both $\hat{\delta}_2$ and $\hat{\delta}_1$ are regular, and we have already shown that this implies that $\hat{\delta}_2 \bullet \hat{\delta}_1(\infty) = 0$.

(iii) For any $K > 0$, $\eta \mapsto \delta(\eta) + \eta\delta(K/\eta)$ is convex on $[0, \infty)$, so it either attains a minimum value or it decreases montonically in $\eta$. But $\delta(\eta) + \eta\delta(K/\eta) \geq \eta\delta(K/\eta) \geq \eta(1-K/\eta)^+ = (\eta - K)^+$, which exceeds $\delta(0) = 1$ for $\eta \geq K + 1$. The minimum over $\eta$ must therefore be attained in $[0, K+1]$. $\qquad\square$

*Proof of Proposition 4.* (i) For any $\eta > 0$, recalling the representation of regular privacy profiles in (45), we have

$$\begin{aligned}
\delta_{Q,R}(\eta) + \eta\delta_{P,Q}(K/\eta) &= \mathbb{E}_Q\left[\left(\frac{dR}{dQ} - \eta\right)^+\right] + \mathbb{E}_P\left[\left(\eta\frac{dQ}{dP} - K\right)^+\right] \\
&= \mathbb{E}_Q\left[\left(\frac{dR}{dQ} - \eta\right)^+\right] + \mathbb{E}_Q\left[\left(\eta - K\frac{dP}{dQ}\right)^+\right]. \\
&\geq \mathbb{E}_Q\left[\left(\frac{dR}{dQ} - K\frac{dP}{dQ}\right)^+\right] \\
&= \mathbb{E}_P\left[\left(\frac{dR}{dQ}\frac{dQ}{dP} - K\right)^+\right] = \mathbb{E}_P\left[\left(\frac{dR}{dP} - K\right)^+\right] = \delta_{P,R}(K),
\end{aligned}$$

(51)

where the inequality (51) follows from $x^+ + y^+ \geq (x+y)^+$. (ii) At $K = 0$, $\delta_{Q,R} \bullet \delta_{P,Q}(0) = 1 = \delta_{P,R}(0)$. For $K > 0$, we will show that equality holds in (51) for suitable choice of $\eta$. Write $G_{Q,R}(x) = 1 + \delta'_{Q,R}(x)$ for the distribution of $dR/dQ$ under $Q$ (as in (10)), and write $G_{Q,P}$ for the distribution of $dP/dQ$ under $Q$. For any cdf $G$, define the inverse
$$G^{-1}(u) = \inf\{x \geq 0 : G(x) \geq u\}.$$

If $dQ/dR$ is an increasing function of $dP/dQ$, then $dR/dQ$ is a decreasing function of $dP/dQ$, and $(dR/dQ, dP/dQ)$ have (by Theorem 2 of Dhaene et al. (2002)) the joint distribution of $(G_{Q,R}^{-1}(U), G_{Q,P}^{-1}(1 - U))$, with $U$ uniformly distributed on $(0, 1)$. Let $\eta_K = \inf\{x : G_{Q,R}(x) \geq 1 - G_{Q,P}(x/K)\}$. By Lemma 1, $G_{Q,R}(x) \uparrow 1$ and $1 - G_{Q,P}(x/K) \downarrow 0$, as $x \uparrow \infty$, so $\eta_K < \infty$. Since $G_{Q,R}(x) \downarrow 0$ and $1 - G_{Q,P}(x/K) \uparrow 1$, as $x \downarrow 0$, we have $\eta_K > 0$. Moreover, the definition of $\eta_K$ implies that

$$G_{Q,R}(\eta_K) \geq 1 - G_{Q,P}(\eta_K/K) \quad\text{and}\quad G_{Q,R}(\eta_K-) \leq 1 - G_{Q,P}(\eta_K/K-).$$

As in Tehranchi (2020), Theorem 2.16, this implies that

$$\{u : G_{Q,R}^{-1}(u) < \eta_K\} \subseteq \{u : G_{Q,P}^{-1}(1-u) \geq \eta_K/K\}$$

and

$$\{u : G_{Q,P}^{-1}(1-u) > \eta_K/K\} \subseteq \{u : G_{Q,R}^{-1}(u) \leq \eta_K\};$$

i.e.,

$$dR/dQ < \eta_K \Rightarrow dP/dQ \geq \eta_K/K$$

and

$$dP/dQ > \eta_K/K \Rightarrow dR/dQ \leq \eta_K.$$

Thus, $(dR/dQ - \eta_K)(\eta_K - K(dP/dQ)) \geq 0$. But $xy \geq 0$ implies $x^+ + y^+ = (x+y)^+$, so equality holds in (51). $\qquad\square$

*Proof of Lemma 6.* We divide the proof into three parts based on the three conditions in Definition 7.

*Equivalence of (a′) and (a):* This follows directly from Theorem 1 and (22).

*Equivalence of (b′) and (b):* If (b) holds, let $\alpha = 1 + \delta'(1) \in (0,1)$. Then, applying (20) we get

$$T_\delta(\alpha) = 1 + \sup_{K \geq 0}\{(\alpha-1)K - \delta(K)\} = 1 + \sup_{K \geq 0}\{\delta'(1)K - \delta(K)\}.$$

The convexity of $\delta$ implies that the supremum is attained at $K = 1$, so

$$T_\delta(\alpha) = 1 + \delta'(1) - \delta(1) = \alpha - \delta(1) \in (0, \alpha),$$

and thus (b′) holds. Conversely, suppose (b′) holds. Then $0 < T(\alpha^*) < \alpha^*$ for some $\alpha^* \in (0,1)$. We will show that $\delta(1) > 0$, $1 + \delta'(1) > \delta(1)$, and $\delta'(1) < 0$, in that order. Applying (18) to evaluate $\delta(1)$ yields

$$\delta(1) = \sup_{\alpha \in [0,1]}\{\alpha - T(\alpha)\} = \alpha^* - T(\alpha^*) > 0,$$

verifying that $\delta(1) > 0$. Again using (18), we have

$$
\begin{aligned}
\delta(1+h) &= -h + \sup_\alpha\{\alpha(1+h) - T(\alpha)\} \\
&\geq -h + \alpha^*(1+h) - T(\alpha^*) \\
&= \delta(1) + (\alpha^* - 1)h.
\end{aligned}
$$

Letting $h \downarrow 0$, we get $\delta'(1) \geq \alpha^* - 1$, and then $1 + \delta'(1) \geq \alpha^* > \alpha^* - T(\alpha^*) = \delta(1)$, as required for (b). Finally, regularity implies $\delta(\infty) = 0$, so $\delta(1) > 0$ entails $\delta'(1) < 0$.

*Equivalence of (c′) and (c):* For $\alpha \in (0,1)$, we have

$$
\begin{aligned}
T_{\delta_0}(\alpha) &= 1 + \sup_{K \geq 0}\{(\alpha-1)K - (1-K)^+\} \\
&= \sup_{K \geq 0}\{\alpha K + \min\{0, 1-K\}\} = \alpha;
\end{aligned}
$$

i.e., $T_{\delta_0} = T_0$. Suppose (c) holds. Then Lemma 9 implies that $T_{\delta_s}(x) \to T_{\delta_0}(x) = x$, for all $x \in [0,1)$. (All tradeoff functions have $T(0) = 0$, so we automatically have $T_{\delta_s}(0) \to T_0(0)$.) At $x = 1$, we have

$$T_{\delta_s}(1) = 1 + \sup_{K \geq 0}\{K \cdot 0 - \delta(K)\} = 1 - \delta_s(\infty) \to 1,$$

where the last step applies (c) at $K = \infty$. Conversely, if (c′) holds, then with $T = T_0$, by using (19) to evaluate $\delta$ we get

$$\delta(K) = 1 - K + \sup_{\alpha \in [0,1]}\{\alpha(K-1)\} = (1-K)^+ = \delta_0(K).$$

Lemma 9 yields $\delta_s(K) \to \delta_0(K)$, for all $K \in [0, \infty)$. Also, $\delta_s(\infty) = 1 + \inf_{K \geq 0}\sup_{\alpha \in [0,1]}\{(\alpha-1)K - T_s(\alpha)\} = 1 - T_s(1) \to 0$; thus, $\delta_s(\infty) \to \delta_0(\infty)$. $\qquad\square$

# D  Omitted Proofs from Section 5

*Proof of Lemma 7.* Differentiability of $f$ implies differentiability of $\Lambda$ and then differentiability of $t$ for $K \in [1, K_{\max}(y))$, $y > 0$, where (36) holds. We may therefore differentiate (37) with respect to $y$ to get

$$
\begin{aligned}
\partial_y \delta(K, y) &= f(y - t(K, y)) - [f(y - t(K, y)) - Kf(-t(K, y))]\partial_y t(K, y) \\
&= f(t(K, y) - y) - [f(t(K, y) - y) - Kf(t(K, y))]\partial_y t(K, y) \\
&= f(t(K, y) - y) \\
&= Kf(t(K, y)),
\end{aligned}
$$

where the second equality uses the fact that $f$ is even, and the third and fourth equalities use the identity

$$
f(t(K, y) - y) = Kf(t(K, y)) \tag{52}
$$

implied by (36). $\qquad \square$

*Proof of Theorem 2.* Rewriting (52) as $f(t(K, y) - y)/f(t(K, y) = K$ and differentiating this ratio with respect to $y$ yields

$$
f'(t(K, y) - y)[\partial_y t(K, y) - 1]f(t(K, y)) - f(t(K, y) - y)f'(t(K, y))[\partial_y t(K, y)] = 0, \tag{53}
$$

so

$$
\partial_y t(K, y)\left[\frac{f'(t(K, y) - y)}{f(t(K, y) - y)} - \frac{f'(t(K, y))}{f(t(K, y))}\right] = \frac{f'(t(K, y) - y)}{f(t(K, y) - y)}.
$$

For log-concave $f$, $f'(z)/f(z)$ is decreasing, so the term in square brackets is positive. It follows that

$$
\partial_y t(K, y) \geq 0 \Leftrightarrow f'(t(K, y) - y) \geq 0.
$$

By (33), $f'(z) > 0$ for $z < 0$ and $f'(z) < 0$ for $z > 0$, so

$$
f'(t(K, y) - y) \geq 0 \Leftrightarrow t(K, y) \leq y.
$$

From the monotonicity of the likelihood ratio in (34),

$$
t(K, y) \leq y \Leftrightarrow \Lambda(t(K, y), y) \leq \Lambda(y, y) \Leftrightarrow K \leq f(0)/f(y) \Leftrightarrow y \geq y_*.
$$

Thus, $t(K, y)$ is decreasing on $(0, y_*)$ and increasing on $(y*, \infty)$. Because $f$ is decreasing on $\mathbb{R}_{++}$, it follows that $f(t(K, y))$ is increasing on $(0, y^*)$ and decreasing on $(y^*, \infty)$. In light of (38), $\partial_y \delta(K, y)$ is increasing on $(0, y^*)$ and decreasing on $(y^*, \infty)$. $\qquad \square$

The proof of Theorem 3 relies on two lemmas.

**Lemma 11.** *The partial derivatives of $t$ satisfy $\partial_y t(K, y)f((t(K, y) - y) = K\partial_K t(K, y)f'(t(K, y) - y)$, for $y > 0$ and $K \in [1, K_{\max}(y))$.*

*Proof.* We can rearrange (53) and apply (52) to get

$$
\partial_y t(K, y) = \frac{f'(t(K, y) - y)}{f'(t(K, y) - y) - Kf'(t(K, y))}; \tag{54}
$$

strict log-concavity of $f$ (i.e., $\psi'' > 0$) ensures that the denominator is not zero. Similarly, differentiating the identity $f(t(K, y) - y) = Kf(t(K, y))$ with respect to $K$ yields

$$
\partial_K t(K, y) = \frac{f(t(K, y))}{f'(t(K, y) - y) - Kf'(t(K, y))} = \frac{f(t(K, y) - y)/K}{f'(t(K, y) - y) - Kf'(t(K, y))}. \tag{55}
$$

Combining (54) and (55) yields the result. $\qquad \square$

**Lemma 12.** *For $y > 0$ and $K \in [1, K_{\max}(y))$, $t(K, y)$ is convex in $y$.*

*Proof.* We will show that $\partial_y t(K, y)$ is increasing in $y$, for $y > 0$ and $K \in [1, K_{\max}(y))$. We will abbreviate $t(K, y)$ to $t$, particularly when it appears as the argument of another function. Divide top and bottom of (54) by $f(t - y) = Kf(t)$ to get

$$\partial_y t(K, y) = \frac{f'(t - y)/f(t - y)}{f'(t - y)/f(t - y) - f'(t)/f(t)},$$

which we can write as

$$\partial_y t(K, y) = \frac{-\psi'(t - y)}{-\psi'(t - y) + \psi'(t)} = \frac{\psi'(t - y)}{\psi'(t - y) - \psi'(t)}. \tag{56}$$

This is increasing in $y$ iff the recriprocal is decreasing in $y$; i.e, iff

$$\frac{\psi'(t)}{\psi'(t - y)} \text{ increasing in } y.$$

Differentiation shows that this holds if the numerator of the derivative is nonnegative; i.e., if

$$\psi''(t)[\partial_y t]\psi'(t - y) - \psi'(t)\psi''(t - y)[\partial_y t - 1] \ge 0,$$

which holds if

$$[\psi''(t)\psi'(t - y) - \psi'(t)\psi''(t - y)]\partial_y t + \psi'(t)\psi''(t - y) \ge 0.$$

Using (56), we may write this condition as

$$[\psi''(t)\psi'(t - y) - \psi'(t)\psi''(t - y)]\frac{\psi'(t - y)}{\psi'(t - y) - \psi'(t)} + \psi'(t)\psi''(t - y) \ge 0.$$

This condition will hold for $t$ and $y$ linked through $t = t(K, y)$ if it holds for all $z, y > 0$, meaning that

$$[\psi''(z)\psi'(z - y) - \psi'(z)\psi''(z - y)]\frac{\psi'(z - y)}{\psi'(z - y) - \psi'(z)} + \psi'(z)\psi''(z - y) \ge 0.$$

This condition holds if (recall that $\psi'$ is increasing because $\psi$ is convex, so $\psi'(z - y) < \psi'(z)$)

$$[\psi''(z)\psi'(z - y) - \psi'(z)\psi''(z - y)]\psi'(z - y) + [\psi'(z - y) - \psi'(z)]\psi'(z)\psi''(z - y) \le 0,$$

which holds if

$$\psi''(z)\psi'(z - y)\psi'(z - y) - \psi'(z)\psi'(z)\psi''(z - y) \le 0,$$

which holds if

$$\frac{\psi''(z)}{[\psi'(z)]^2} \le \frac{\psi''(z - y)}{[\psi'(z - y)]^2}$$

which follows from (39). $\qquad \square$

*Proof of Theorem 3.* In light of Lemma 8, it suffices to show that the right side of (41) is increasing in $y$, or, equivalently that its logarithm is increasing in $y$. For $1 \le K_1 < K_2 \le K_{\max}(y)$, we have

$$
\begin{aligned}
\log f(t(K_2, y) - y) - \log f(t(K_1, y) - y) &= \int_{K_1}^{K_2} \partial_K \log f(t(K, y) - y) \, dK \\
&= \int_{K_1}^{K_2} \frac{f'(t(K, y) - y)}{f(t(K, y) - y)} \partial_K t(K, y) \, dK \\
&= \int_{K_1}^{K_2} \frac{1}{K} \partial_y t(K, y) \, dK,
\end{aligned}
$$

where the last step uses Lemma 11. Lemma 12 implies that the integrand is increasing in $y$ and therefore that the integral is increasing in $y$. $\qquad \square$

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
