# OpenReview forum: "Privacy Profiles Under Tradeoff Composition"
_TMLR — Accepted by TMLR_

### Review · Reviewer_NdRK · 2025-11-07

**Summary Of Contributions:**

This work studies the connections between the two frameworks used for comparing differential privacy guarantees.
Starting from the definition of T-convolution, the authors establish the closure and composition properties of privacy profiles and show, through convex conjugacy with tradeoff functions, several claims can be made; some of them correspond to the existing research results and some are new.
Also, the authors propose the concept of regularity for privacy profiles and connect them to tradeoff functions as well.

**Additional Comments:**

I recommend acceptance after revising the minor parts mentioned above.

**Audience:**

Yes

**Audience Explanation:**

The established connection with T-convolution, regularity etc. is elegant.
If I raise one thing, the manuscript is not self-contained.  So for readers who are not familiar with several related field may find it difficult to grasp the claims clearly.

**Claims And Evidence:**

Yes

**Claims Explanation:**

The paper is well-written; the math seems rigorous and correct.

**Requested Changes:**

Only some minor corrections:
1. Better to add definition of ( )^+ just in case
2. (15) and (29); do they take y as input or not; need to be consistent
3. In page 14; typo “worst-case bound in” → “worst-case bound is”
4. Better not to refer to something in appendix in the main text.

---

> ### Author Response · Authors · 2025-12-18
> **Corrections made**
>
> Thank you for your review and comments!
>
> To make the paper more self-contained, we have added Appendix A, which provides some technical background results. We have also added more explanation of several results in the main body of the paper.
>
> - We have added the definition of ()^+ just before Definition 3, where it is first used
> - Just before what is now (24) (the old (29)), we now explain to the reader the modification in notation. We need the second argument in (24) but it would create other inconsistencies if we used it earlier, where \delta always has a single argument.
> - We have corrected the typo
> - We have removed from the main text all references to equation numbers in the appendix. We do in several places refer to Appendix A parenthetically for supporting technical details. This technical appendix was added in response to suggestions from another reviewer. But the main text does not rely on these references.

---

> > ### Comment · Reviewer_NdRK · 2025-12-19
> > **Thank you for the response**
> >
> > Thank you for the response; it is satisfactory.

---

### Review · Reviewer_AM2J · 2025-11-18

**Summary Of Contributions:**

This paper focuses on privacy profiles and tradeoff functions, two frameworks to analyze differential privacy mechanisms and comparing their privacy guarantees. The authors begin by discussing the connection between privacy profiles and tradeoff functions. Subsequently, they introduce the notion of regular privacy profiles and some related theoretical results. This discussion sets the stage for the description of T-convolution of privacy profiles, which is shown to correspond to the composition of tradeoff functions. This is clearly distinguished from composition of privacy mechanisms. The authors conclude their work by analyzing the convexity properties of privacy profiles for log-concave additive noise, with examples such as the noise added in the Gaussian mechanism.

The central ideas and theoretical results presented in this paper constitute key contributions to the space of differential privacy. In particular, the introduction of T-convolution, and its correspondence to the composition of tradeoff functions is itself a noteworthy contribution to this space. I also appreciate the authors’ thoroughness and rigor in proving the various propositions, lemmas and theorems delineated in this work.

However, this rigor comes at the expense of clarity. I oft found it a struggle to keep track of the “bigger picture” in each section as the authors sought to prove each result right after stating it. In other instances, there are entire paragraphs filled with mathematical expressions that are hard to follow. All of these detract from the readability of the paper, and distract the reader from the key results in this work. In a similar vein, there are references made to theoretical results from many references in the proofs; a restatement or summary of some of these key results would also improve the readability of the manuscript.

**Additional Comments:**

**Questions:**

1.	What is the purpose of the discussion included in the last paragraph preceding Section 3.2? Does this have any implications for any of the subsequent results? If not, this could be removed as it seems to suggest that the idea of “dominating a pair of distributions” is problematic.
2.	How did you arrive at the first expression in the proof of (23)?

**Audience:**

Yes

**Audience Explanation:**

The authors have been very thorough and rigorous in proving all the theoretical results included in this paper, building upon results from existing literature. Specifically, the introduction of the binary operation of T-convolution of two privacy profiles and its equivalence to the composition of the respective tradeoff functions serves as an advances a hitherto unseen perspective on privacy mechanisms. The subsequent discussion of its applications to the convexity of log-concave additive noise may also be valuable to readers working in the space of differential privacy.

**Broader Impact Concerns:**

I do not have any broader impact concerns with this work.

**Claims And Evidence:**

Yes

**Claims Explanation:**

The authors are thorough and rigorous in proving all the theoretical results included in the manuscript, sometimes at the expense of clarity. While I was unable to check all the proofs, the ones that I did appeared to be valid.

**Requested Changes:**

**Critical changes:**

1.	I strongly urge that the authors only retain the proofs of only the key results in the main paper, and instead focusing on guiding the reader towards the key findings/intuitions of the paper, such as the implications of T-convolution and its presentation as a composition of the respective tradeoff functions. The other proofs can be moved to the appendix.
2.	Where possible, please restate or summarize the key results from existing literature, especially if used in the proofs. This may be done in the appendix.

**Suggested changes:**

1.	Page 3: “convient”  “convenient”
2.	Page 3: “with respect P”  “with respect to P”
3.	There is a paragraph on page 16 preceding Section 5, that is full of mathematical expressions. There may be a less dense and clearer way of presenting this information.

---

> ### Author Response · Authors · 2025-12-18
> **Changes made**
>
> Thank you for the review and comments!
>
> As suggested, we have moved almost all proofs to the appendix, and we agree that this significantly improves readability of the paper. To help guide the reader, we have added language in several places (including following Proposition 1, following Proposition 2, beginning of Section 5, expanded Remark 1, and others) to provide more context and insight for the results.
>
> In response to the second comment, we have added a new Appendix A, which explicitly states several technical results used in the paper, some of which may be less familiar to readers. In the main body of the paper, we have provided explicit references or referred to Appendix A when we use a result from the existing literature.
>
> We have corrected the two typos. Regarding the paragraph previously on p.16, we agree; this discussion now appears at the end of Appendix C (in the proof of Lemma 6), where we have broken the argument into three parts and provided more of a step-by-step explanation. The previous paragraph is now spread over nearly a page and should be much more intelligible.
>
> Regarding the paragraph that previously came just before Section 3.2, we agree that it was unnecessary and have removed it as suggested. For the second question, the first expression comes from Dong et al., and we have added an explicit reference to their Proposition 6. The expression for $T^*_o$ comes from the definition of the convex conjugate, about which we have included further background in the new Appendix A.

---

### Review · Reviewer_NdCd · 2025-12-13

**Summary Of Contributions:**

This paper connects two key notions used to characterize the differential privacy of mechanisms: the privacy profile (Definition 2) characterized by the function $\delta_{Q, P}$ and tradeoff functions (Definition 4) characterized by $T(P, Q)(\alpha)$. The paper focuses mainly on mutually absolutely continuous random variables (because otherwise we'd probably have large $\delta$) and defines the notion of a "regular" function $\delta$ (Defn. 5) and show that a function $\delta$ is regular if and only if it is the privacy profile of a pair of mutually absolutely continuous random variables (Proposition 1). Given a regular function $\delta$, we can further say that it corresponds a symmetric privacy profile (one for whom $\delta_{P, Q} = \delta_{Q, P}$ for $P, Q$ mutually absolutely continuous random variables) if it satisfies $\delta = \hat{\delta}$ for $\hat{\delta}$ defined by Equation (19). Proposition 2 then shows this $\hat{\delta}$ behaves as expected: when $\delta = \hat{\delta}$ then $\delta_{P, Q} = \delta_{Q, P}$. Building towards the main result of the paper, Lemma 2 shows a 1-to-1 connection between tradeoff functions and privacy profiles for the same pair $P, Q$ obtained through the convex conjugate. This allows us the authors to characterize regular privacy profiles in terms of their tradeoff functions (Proposition 3). Theorem 1 shows that the composition of tradeoff functions corresponds to the "T-convolution" of privacy profiles defined in Defn. 6. The composition of tradeoff functions shows up in many places in DP and thus connecting it to what happens to the corresponding privacy profiles is a good contribution. An example of one such place is group privacy, where datasets can differ by $m$ points rather than $1$ (or whatever the original number of points they differ by in the equivalence relation). Section 4.3 shows that if $\delta$ bounds privacy for neighboring databases, then the group privacy profile is lower bounded by the $m$-fold T-convolution $\delta^{\bullet m}$, and Lemma 5 implies $\delta^{\bullet m}$ is at least as large as the original privacy profile, as expected. The paper recovers known group privacy bounds (e.g., Vadhan-type scaling) and connects them to the T-convolution.  The rest of the paper derives some more properties of the privacy profiles of log-concave additive noise mechanisms, with the main result showing that for such profiles the T-convolution corresponds to adding sensitivity parameters (which is tight in the case of e.g. Gaussians).  Overall, while this paper does not introduce any new mechanisms or algorithms, it helps to uncover some deep connections between privacy profiles and tradeoff functions and allows for easier navigation of existing literature using either of these two notions to characterize DP.

**Additional Comments:**

I apologize for the very long delay in submitting my review.

**Audience:**

Yes

**Audience Explanation:**

Yes, this paper connects privacy profiles / privacy curves and tradeoff functions in a way that is useful to studying and understanding group privacy and additive-noise mechanism. This connection is very useful for making sense of a literature that often uses one or the other. The paper gives a clean algebraic bridge (the T-convolution) that lets results transfer between the profile and tradeoff frameworks, plus some new structural properties for log-concave mechanisms. While the T-convolution is not a novel contribution of this paper (coming from [1]), its application to differential privacy seems to be.

[1] Michael R Tehranchi. A black–scholes inequality: applications and generalisations. Finance and Stochastics,
24(1):1–38, 2020.

**Broader Impact Concerns:**

N/A.

**Claims And Evidence:**

Yes

**Claims Explanation:**

The proofs look correct and the definitions introduced are well-motivated (primarily by the resulting connections). The connection shown between tradeoff functions and privacy profiles is clear and has several practical implications (shown in Sections 4.3 and 4.4). The results shown in Sections 4.3 and 4.4 are also shown to be essentially tight: for group privacy, the paper gives a construction showing the $\delta^{\bullet m}$ bound can be attained in a worst-case sense, and for log-concave additive noise mechanisms the sensitivity-additivity statement is tight for standard examples (e.g., Gaussian). This answers the question of whether we can use the T-convolution to obtain tight results essentially positively, though it also suggests we should not expect the T-convolution alone to systematically improve upon existing bounds in more specific cases.

**Requested Changes:**

I have no requested changes and recommend the paper for acceptance as it is.

---

> ### Author Response · Authors · 2025-12-18
> **New version**
>
> Thank you for your review and comments!

---

### Author Response · Authors · 2026-01-29
**Minor revision**

We appreciate these thoughtful and constructive comments -- thank you. They are incorporated in the new version of the paper.

- Lemma 2: The version of Prop 6 of Dong et al. (2022) that we need is stated as Lemma 20 of Zhu et al. (2022). That version is stated for a (P,Q) pair rather than a mechanism, without assuming symmetry, and for all $K\ge 0$ (their $\alpha \ge 0$). Since Zhu et al. did not provide a proof, we have added Lemma 10 in Appendix C, which proves the result we need. Following the proof of the lemma, we have added some discussion of how the symmetric and asymmetric results are related.

- Definition of tradeoff function: Our definition corrects a minor typo (a flipped inequality) in Awan and Dong (2022), which is also noted in Awan and Ramasethu (2024). We now point this out immediately before Definition 4.

- $\delta$ notation: Our convention makes $\delta_{P,Q}$ the (shifted) conjugate of $T_{P,Q}$ rather than of $T_{Q,P}$. This allows us to suppress the $(P,Q)$ in, e.g., (18) and (19), and just refer to $T$ and $\delta$ as the tradeoff function and privacy profile for $(P,Q)$. Otherwise, we would have to refer to the privacy profile for $(Q,P)$ and the tradeoff function for $(P,Q)$. We have added a footnote on p.4 to make this point.

- Argument of $\delta$. We have added a statement in the paragraph that follows Definition 2 emphasizing that unless otherwise stated, $\delta$ is a function of $K\ge 0$. In the randomized response example, we have changed $\delta(\epsilon)$ to $\delta(e^{\epsilon})$. We only use $\epsilon$ where it is needed to make a connection to other work.

- Derivative of $\delta$. We have added two sentences just before Definition 5 explaining that $\delta$ is at least right-differentiable even in the case of a discrete likelihood ratio. In the discrete case, $\delta$ is piecewise linear and its derivative is piecewise constant. These properties become clear from the representation in (9), which expresses the privacy profile as the tail integral of the complementary cdf of the likelihood ratio.

- Typos corrected

- Computation: We have added a comment just before the Gaussian example on p.10. Equation (5) of Dong et al. and the relevant result from Wasserman and Zhou (2010) are referenced following the new Lemma 10 in Appendix C.

- We see the results in Section 4.4 as helping to connect properties of privacy profiles in the paper with other results in the literature, particularly for privacy profiles.

---

> ### Comment · Action_Editor_FZ5s · 2026-01-30
>
> This clarifies everything. Thanks a lot and congratulations on a really nice paper!

---

### Decision · Action_Editor_FZ5s · 2026-01-21

**Recommendation:** Accept with minor revision

**Additional Comments:**

All reviewers evaluated the paper positively. The authors have adequately addressed the reviewers’ comments in their rebuttal and revised manuscript.

I would also like to thank Matthew Regehr (University of Waterloo) for helpful comments on the paper.

----

The paper seems to be largely mature; however, I am requesting minor revisions related to Lemma 2 and the definition of the trade-off function:

Instead of using the "usual" trade-of function as defined by Dong et al. (2022) (https://arxiv.org/pdf/1905.02383v1), the horizontally flipped trade-off function is used here, meaning $T(\alpha) = T_o(1-\alpha)$, where $T_o$ is the trade-off function defined by Dong et al. (2022).  Then, Prop. 6 by Dong et al. (2022) is used for $T_o$. There is a statement "They assume $T_o$ is symmetric but the result holds even without this condition", however there is no formal result to back this up. I can see this follows from the proof given in (Dong et al., 2022), however perhaps it would be good to clarify or give some formal statement for this step? I see also Zhu et al. (2022) (https://proceedings.mlr.press/v151/zhu22c/zhu22c.pdf) use this relation between the hockey-stick divergence and trade-off function in their Lemma 21 (which does not require the trade-off function to be symmetric), however they also refer to the result by Dont et al. (2022) without giving a proof for it.

Related: when defining the trade-off function, you write "We turn next to the definition of tradeoff functions as introduced in Dong et al. (2022) and modified in Awan & Dong (2022)", however it looks to me that the definition given by Awan and Dong (2022)(https://proceedings.neurips.cc/paper_files/paper/2022/file/dd73933d99ccd7ffe2306adb95ec5d02-Paper-Conference.pdf) is not the same as the one used here. There are now three different definitions:

Dong et al. (2022):

$$
T_o(P, Q)(\alpha) = \inf_{\phi} \\{ 1 - \mathbb{E}_Q [\phi]  :  \mathbb{E}_P[\phi] \le \alpha \\}, \alpha \in [0,1].
$$

Yours:

$$
T(P, Q)(\alpha) = \inf_{\phi} \\{ 1 - \mathbb{E}_Q [\phi]  :  \mathbb{E}_P[\phi] \le 1-\alpha \\}, \alpha \in [0,1].
$$

which is indeed the $x$-axis-flipped trade-off function of Dong et al. (2022), and

Awan and Dong (2022):
$$
T_{AD}(P, Q)(\alpha) = \inf_{\phi} \\{ 1 - \mathbb{E}_Q [\phi]  :  \mathbb{E}_P[\phi] \ge 1-\alpha \\}, \alpha \in [0,1].
$$

If possible, could you comment on these differences, or at least change the wording when arriving at your definition?

Also, the privacy privacy profile $\delta_{(P,Q)}(K)$ seems to be defined in non-traditional way by flipping $P$ and $Q$. Is there some specific motivation for using these definitions for the privacy profile and trade-off function?


----

Few minor things:

- Some confusion with notation delta: mostly $\delta(x) = h(x)$ but in the randomized response example $\delta(x) = h(\exp(x))$ where $h(x)$ denotes the privacy profile parametrized with $x \in [0,\infty)$

- The meaning of the derivative of delta not always clear. For example, if we have a discrete-valued privacy loss random variable, is $\delta'(x)$ then well-defined via the right derivative that is used here? Even if yes, I think one or two clarifying sentences would be helpful when it is first introduced.

- There are few typos: "mechansm", "T-convoluation", "beween", "liklihood", "requring"
----

Overall, the paper is of high quality and will be a valuable addition to the DP literature. That said, I think the contributions are more consolidating than breakthrough in their nature. For instance:

- As mentioned above, I find that one slight deficit with the stated T-convolution formulas is that they don't seem to be efficiently computable numerically. Or, at least, I do not see how they would be "more efficiently" computable than compositions of trade-off functions. One formula the paper seems to be missing is the direct translation between trade-off functions and privacy profiles given by Wasserman and Zhou (2010), as given in Eq. 5 by Dong et al. (https://arxiv.org/pdf/1905.02383). Instead of numerically evaluating T-convolutions, one could numerically compose trade-off functions and translate them to privacy profiles, to obtain group privacy guarantees.

- Although the mathematics is throughout interesting, some results feel like having limited added value, like the T-convolution result for additive noise mechanisms given in Subsection 4.4, for example. Prop. 7 shows that the sensitivities add up for T-convolution of two privacy profiles corresponding to mechanisms with identical additive noise distributions. One the other hand, it is trivial that if the underlying function $f(X)$ ($X$ denoting a dataset) to be "blurred" has sensitivity $y$, then $ || f(X)-f(X') || \leq k y$ in case $X$ and $X'$ differ in $k$ data elements. And also one easily sees this bound is tight. So the T-convolution gives a mathematical tool which has these more general properties, however for group privacy simpler approach would likely suffice.

**Audience:**

Yes

**Audience Explanation:**

This is a very interesting paper for anyone interested in theoretical aspects of differential privacy, meaning, to quite a large subset of TMLR's audience. The main contribution of the paper, the T-convolution of the privacy profiles to get group privacy guarantees is really interesting. Also, the connection between financial mathematics and various aspects of DP such as the hockey-stick divergence and Gaussian mechanism are very interesting. There are also several smaller results, mathematical properties of trade-off functions and privacy profiles stated, which are not equally significant but many of which seem to be missing from the literature. One slight deficit with the main result is that there is no directly computable expression for T-convolutions, like, for example, in the expression for compositions of mechanisms where the FFT-computable convolutions of privacy loss distributions appear.

**Claims And Evidence:**

Yes

**Claims Explanation:**

This is a theory paper and of high quality. The claims are backed by mathematical proofs. There are few minor issues which I believe are fixable, see below for more details.